# The Effect of Cleat Position on Running Using Acceleration-Derived Data in the Context of Triathlons

**DOI:** 10.3390/s21175899

**Published:** 2021-09-02

**Authors:** Stuart A. Evans, Daniel A. James, David Rowlands, James B. Lee

**Affiliations:** 1SABEL Labs, College of Health and Human Science, Charles Darwin University, Darwin, NT 0810, Australia; dan@qsportstechnology.com (D.A.J.); Jim.Lee@cdu.edu.au (J.B.L.); 2School of Engineering, Griffith University, Brisbane, QLD 4111, Australia; d.rowlands@griffith.edu.au

**Keywords:** accelerometer, sensor, centre of mass, cycling, kinematics

## Abstract

Appropriate cycling cleat adjustment could improve triathlon performance in both cycling and running. Prior recommendations regarding cleat adjustment have comprised aligning the first metatarsal head above the pedal spindle or somewhat forward. However, contemporary research has questioned this approach in triathlons due to the need to run immediately after cycling. Subsequently, moving the pedal cleat posteriorly could be more appropriate. This study evaluated the effectiveness of a triaxial accelerometer to determine acceleration magnitudes of the trunk in outdoor cycling in two different bicycle cleat positions and the consequential impact on trunk acceleration during running. Seven recreational triathletes performed a 20 km cycle and a 5 km run using their own triathlon bicycle complete with aerodynamic bars and gearing. Interpretation of data was evaluated based on cadence changes whilst triathletes cycled in an aerodynamic position in two cleat positions immediately followed by a self-paced overground run. The evaluation of accelerometer-derived data within a characteristic overground setting suggests a significant increase in total trunk acceleration magnitude during cycling with a posterior cleat with significant increases to longitudinal acceleration (*p* = 0.04) despite a small effect (*d =* 0.2) to the ratings of perceived exertion (RPE). Cycling with a posterior cleat significantly reduced longitudinal trunk acceleration in running and overall acceleration magnitudes (*p* < 0.0001) with a large effect size (*d* = 0.9) and a significant reduction in RPE (*p* = 0.02). In addition, running after cycling in a posterior cleat was faster compared to running after cycling in a standard cleat location. Practically, the magnitude of trunk acceleration during cycling in a posterior cleat position as well as running after posterior cleat cycling differed from that when cycling in the fore-aft position followed by running. Therefore, the notion that running varies after cycling is not merely an individual athlete’s perception, but a valid observation that can be modified when cleat position is altered. Training specifically with a posterior cleat in cycling might improve running performance when trunk accelerations are analysed.

## 1. Introduction

Multisport events such as triathlons commonly involve consecutive cycling and running phases. Overall performance in these events is highly correlated with economy of motion in the sub-discipline of changing from cycling to running [1,2,3]. Reports from athletes involved in triathlons indicate that one of the most difficult parts of a race can be the transition from the non-weight bearing activity of cycling to the weight-bearing task of running [4]. This is due to the combined physiological, biomechanical and neuromuscular challenges associated with the cycle-to-run segment [5,6]. In this regard, studies have shown that for overall race success in a triathlon, one must effectively and efficiently transition from cycling to running. This transition can be critical in determining overall race placement, particularly in shorter duration triathlons [4].

Trunk position is known to affect pedalling frequency, torque delivery and effectiveness in cycling [7] as well as vertical oscillation in running [8]. Vertical oscillation can be described as the amount of displacement that the body’s centre of mass (COM) experiences (i.e., the up and down motion of the COM). Since a substantive proportion of the race duration is spent cycling and running, it is important to recognise the significance of trunk position given the abrupt change from a near horizontal trunk profile in cycling to a vertical trunk in running. For instance, trunk angle affects leg kinematics, related muscle activity and limb length [7], with a forward shift of the COM meaning that the athlete is less supported by the saddle [9]. It could be argued that a lesser COM displacement may also affect peak/mean power given the power−cadence relationship. This raises questions on both the limiting factors as well as the significant relationship between the body COM in cycling and running. Thus, strategies that lessen the kinematical magnitude of trunk motion during cycling may prove advantageous in enhancing run performance.

Numerous factors have been identified to play an important role for running performance in triathlons, including cycling pacing strategy [10], pedalling cadence [11], position of the triathlete whilst cycling [12] and cleat position [13,14,15]. The latter has been the focus of contemporary research in that cleat location can optimise lower limb kinematics throughout cycling which could improve performance [16]. Despite minor variations, the threading to accept cleat hardware is consistently located underneath an anatomical landmark considered to be the most appropriate, that is, in the region of the third metatarsophalangeal joint (MPJ). In this instance, the first metatarsal head is positioned directly above the pedal spindle (i.e., fore-aft) [17,18]. However, despite this location being suggested to be the most optimal for performance, it is controversial [19], not well investigated and may promote a higher likelihood for injury [20]. Whereas the MPJ region appears to have been adopted by a significant array of cyclists, it remains largely theoretical whether the location is beneficial for triathletes given the run segment that follows. Therefore, determination of alternative cleat locations could potentially reduce injury while improving both cycling and running performance. Along this line, recent studies advise placing the cleat slightly backward (posteriorly), in alignment with the middle of the first and fifth metatarsal heads [16,21] (Figure 1). Research suggests that placing the cleat posteriorly (POS) decreases the load on the lower limbs [15], possibly reducing plantar flexor muscle activity during cycling, which may be of benefit during subsequent running [22].

Two studies have investigated the effects of a mid-foot cleat position on biomechanical variables during both cycling and running of a simulated sprint distance run. Millour et al. [14] evaluated the impact of fore-aft and POS cleat position on biomechanical and physiological variables during the cycling and running parts of a simulated sprint triathlon. The authors concluded that the POS cleat position could have practical benefits for subsequent running and could be recommended for use in triathlons when running 5 km and over. Paton and Jardine [4] showed that a mid-foot cleat position improves the subsequent performance of a 5.5 km time trial running exercise after 30 min of cycling at 65% of the mean arterial pressure (MAP). These results appear to be supported by Litzenberger et al. [21] who found that a posterior cleat decreases calf-muscle activity by reducing the lever arm of the foot and, consequently, ankle joint torque. By changing the location of the cleat, the relative position of the trunk changes, possibly making the lower-limb muscles work at different portions of their force-length relationships. Nevertheless, contrary to Paton and Jardine [4], Viker and Richardson [15] found no significant performance improvement during a 5 km time-trial running exercise if the 30-min cycling part was performed at a non-constant power output between 50% and 90% of the MAP. Nevertheless, the pacing strategy during the running test was significantly modified as the first kilometre was performed more slowly when running from the mid-foot cleat position compared to using the traditional metatarsal cleat position. Although these two previous studies provide some interesting findings, temporal trunk kinematics were not included in the cycling and running exercise.

Besides reducing lever arm length when cycling with a posterior cleat, the effect of trunk angle on muscle recruitment patterns is a factor in performance. Savelberg et al. [7] revealed that trunk angle affects the kinematics of leg movement and muscle activity, not only at the hip, but at the knee and ankle alongside muscles spanning these joints. In this respect, the effect of trunk position has a bearing on overall performance when in a closed kinetic chain. Furthermore, the authors suggested that trunk angle influences power output, whilst Wille et al. [8] found that both cadence and the magnitude of COM excursion are associated with vertical ground reaction forces and a braking impulse in running. Thus, instructing a triathlete to reduce acceleration of the trunk may be a viable alternative to traditional methods of interventions.

While cycle cleat position and subsequent running performance have been studied quite extensively, relatively little is known about the relationship between trunk acceleration magnitude and cleat position, which may explain some of the detriment previously observed in running, particularly vertical oscillation (longitudinal acceleration). Advances in sensor technology provide clinicians with the capability to assess cycling and running kinematics external to the traditional laboratory environment. The use of sensor technology demonstrates how widely available mobile technology can be used to quantify kinematics without the need for a fully instrumented gait laboratory [22]. Consequently, the continuous collection of data recorded by a sensor allows for regular analysis during a cycle and run in ‘real time’ which subsequently provides information about what and when changes to trunk position occur. By studying, in detail, the relationship between trunk acceleration, cleat position, cadence and the consequential effect on running, more insight may be obtained into how these variables are related. Consequently, the purpose of this study was twofold. First, to characterise the change in trunk acceleration magnitude in triathletes instructed to cycle with an MPJ cleat position and posterior (POS) cleat position. Second, to determine if a POS cleat is more beneficial due to lower magnitudes of longitudinal oscillation when running from cycling.

## 2. Materials and Methods

This study consisted of seven recreational triathletes (age: 40 ± 10 years, height: 173 ± 8.9 cm, weight: 71 ± 8.1 kg, weekly training frequency: 9 ± 2 h, saddle height: 79 ± 0.6 cm, inseam: 76 ± 3 cm, seat tube angle (STA): 79° ± 0.69), recruited by word of mouth and within the local triathlon community. All participants were healthy at the time of testing and had no acknowledged neuromuscular or musculoskeletal disorders at the time of the study. The participants were asymptomatic of illness and free from any acute or chronic injury, as established by the American College of Sports Medicine [23] participant activity readiness questionnaire (PAR-Q) with a protocol approved by the University’s Research Ethics Committee (HREC 030317). Individuals used their own bicycles with integrated aerodynamic bars and gears as is common on triathlon bicycles. Participants were asked to refrain from vigorous training 24 h prior to both tests and were instructed to preserve their typical diet. Participants were tested by means of their own bicycles to eliminate the effects of unfamiliarity.

### 2.1. Methodology

Cycling and running kinematic data were assessed by means of triaxial accelerometer outputs. Specifically, trunk acceleration magnitude changes to cycling cadence in both cleat positions and acceleration magnitudes in running after cycling were measured as m/s². Data for each cycling variable (cadence) and cleat position were registered during the overground cycle and were calculated as a 60 s mean of each 5 km lap of the overground bicycle course. The purpose of this was to ensure that a stable pacing strategy and cadence stabilisation was attained. Next, to analyse conceivable kinematical variation between MPJ and POS cleat positions during each 5 km lap of cycling and corresponding cadence, mean data were stratified into x, y, z components. Ensuing triaxial acceleration of the trunk in running was calculated as the average magnitude for each 1 km epoch of running. Mean running data were also stratified into x, y, z components relative to each 1 km epoch of running.

Triathletes include workouts in their training plans that stack two disciplines, one after the other, with minimal to no breaks in between. This is because one of the most important aspects of this sport is the transition from cycling to running, which is a key factor in achieving a good result [1,2]. Therefore, to replicate a typical training condition and to accomplish the purpose of this study, participants cycled at a varied yet progressively augmented cadence for 20 km followed by a 5 km overground run whilst wearing a triaxial accelerometer. A total of two outside experiments were performed. Experiment 1 (day 1) required participants to complete a 4 × 5 km (i.e., 20 km) loop of overground cycling performed at varied yet progressively increased cadence ranges whilst cycling in an accustomed (MPJ, fore-aft) shoe cleat position. This was followed by a 5 km overground run performed at self-selected pace. The next experiment (day 2) required participants to complete the same 4 × 5 km loop (20 km) of overground cycling that was performed at the same varied yet progressively increased cadence ranges in an adjusted cleat (POS) position. This was followed by a 5 km overground which was once again performed at self-selected pace. A period of approximately 60 s was permitted after participants completed the 20 km cycle route in both experiments in order for participants to change from cycling footwear into running footwear. Both experiments followed the same protocol as stated in Table 1.

Participants wore a standard one-piece triathlon racing suit and were evaluated at the same time of the day (between 0600–0900) under similar environmental conditions (20–22 °C, 70–75% relative humidity). These times were knowingly designated due to the overground cycle route being free from interference, such as vehicles. A duration of 7 days separated experiment 1 (day 1) and experiment 2 (day 2) in order to reduce the possible influence of learning effects. Participants were not provided with the cadence conditions between this period and were not informed if the same cadences would be used during both experiments.

A predominately flat overground asphalt route was selected for the cycling component of both experiments. The course is frequently used by triathletes for time trial (TT) performance [24]. Participants started and finished cycling and running at site 1 as highlighted in Figure 2. The asphalt route was knowingly selected in order to restrict braking whilst cycling as this mimics the cycling conditions during triathlon. Furthermore, the route was deemed to have a low level of technical difficulty which required negligible reason to brake. Consequently, the authors considered this a practical and viable route that mimics a typical sprint distance triathlon and permitted for appropriate evaluation of sensor output comparative to realistic performance application.

Site 1 signified the commencement of each 5 km lap of cycling and subsequent change of cadence whilst representing the start and finish position for the proceeding run. Cadence changes were verbally communicated to participants once they approached the site. To signify the completion of one 5 km lap and cadence condition, the sensor was manually synchronised by the authors as participants rode past the demarcated site in order to categorise synchronisation points in the raw data through post hoc analysis. Cadence was selected in a manner consistent with a previously established protocol [25] and was nominated owing to its ease of measurement in that all participants had fitted speedometers. Prior to cycling, speedometers were manually tested by the principal author to ensure accurate reading. Cadence was viewable via the individual display meters so that participants could monitor the appropriate rev/min¹. Participants performed both cycling experiments in their accustomed aerodynamic position. This position is defined as elbows on the pads of the aero-handlebars with elbow angle close to 90° and the upper part of the trunk parallel to the ground [26].

The bicycles used in the study were classified as triathlon, or time-trial bicycles, that comprised steeper seat tube angles compared to road bicycles. This has an effect on the triathlete′s position, generally bringing the rider further forward over the bottom bracket. Apart from standard group-set parts such as brake callipers and chain sets, which are also common on road bicycles, the fixed aerobars, lightweight brake levers and bar-end gear shifters found on triathlon bicycles allow the triathlete to essentially cycle with one hand and arm placement and, therefore, one riding position. In this instance, the brake levers are located posteriorly to the aerobars and require a deliberate change in hand and therefore trunk position to either reach for the front and/or rear brake. This is in contrast to a road bicycle whereby the brake levers are located directly in front of the drop bars, thereby staying relatively close to the intended cycling position. In this sense, the aerodynamic position generally places more reliance on using the integrated gearing shifters located at the end of the aerodynamic bars, which differs from that used by road cyclists. Therefore, triathletes are likely to “shift up or down” to a lesser gearing ratio to reach a self-selected and efficient cadence to preserve performance [24]. Additionally, the aerodynamic position is frequently used by triathletes in both training and competitive environments as a means to reduce their frontal projected profile in order to minimise the effects of aerodynamic drag.

### 2.2. Instrumentation and Measurement

An inertial measurement unit (IMU), specifically a triaxial accelerometer (52 mm × 30 mm × 12 mm, mass 23 g; resolution 16-bit, full-scale range 16 g, sampling at 100 Hz: SABEL Labs, Darwin, Australia) was fixed to participants′ spinous process (L5/S1) using double-sided elastic adhesive tape for measurement purposes. Explicitly, linear accelerations at the sensor were measured on the skin over spinous processes, defined as the lumbar vertebrae position 5 (L5) and sacrum vertebrae position 1 (S1). This location was selected as it is the unique and closest external point to trunk movements and the point of distribution of the weighted position vectors (net force) that sum to zero [24]. Throughout cycling, movement of the lower limbs in the sagittal plane was generally constrained to a circular path by the geometry of the bicycle (i.e., by crank length and pedals) and body position of the triathlete on the bicycle. Therefore, within these limitations, the triathlete can vary pedalling technique by changing the kinematics of their upper body lower limbs; consequently, this change can be detected by the accelerometer. Subsequently, if a triathlete has undesirable trunk movement when cycling (i.e., varying mediolateral trunk motion when the direction of travel is linear), the acceleration of that movement can be sensed, stored on the local device and analysed. A static calibration was performed on all sensors according to Lai et al. [27] prior to commencing cycling. The same calibration process was repeated for experiment 1 (day 1) and experiment 2 (day two). This process also served to check channel orientations associated to each axis of interest [28,29]. Calibration was performed in accordance with the manufacturer’s specifications (SABEL Labs). The device hardware specifications included a ±2 gravitational acceleration (g), ±4 g, ±8 g and ±16 g selectable scale. Participants were able to cycle freely overground and outside of the traditional laboratory environment due to data being stored locally on the IMU. The IMU was controlled remotely (wirelessly) from the principal author via a typical Hewlett Packard laptop computer using an all-inclusive MATLAB Toolkit. This permitted for control of multiple IMUs which allowed no restrictions during cycling and running data capture. Data was subsequently downloaded from the IMU using a SABEL Sense software program (SABEL Sense 1.2_x64, SABEL Labs) via a CSV file before being converted to a macro-enabled Microsoft Excel workbook. All sensors were powered by a single cell Li-Ion battery and were positioned to measure trunk acceleration data in three orthogonal planes where longitudinal (LN), mediolateral (ML) and anteroposterior (AP) aligned with x, y and z, respectively (Figure 3).

Raw sensor data was then scaled into metres per second/per second (m/s^2^), as is frequently seen in sport science literature [28,29]. Filtering was not applied to the sensor data. As the trunk undergoes movement in both cycling and running, the magnitude of trunk acceleration, as detected at the spinous process, will be a function of its local x, y and z acceleration components. In this respect a postural change will be apparent in the local acceleration components. In this paper, trunk accelerations of each local component were collected for each participant to examine the longitudinal, mediolateral and anteroposterior changes in both cleat positions (i.e., experiment 1 and experiment 2) and the consequential impact on running after cycling. To avoid any effect of fatigue, learning effect or drift, triaxial trunk acceleration was analysed for 60 s at the end of each cycle lap and cadence range, excluding the initial warm up period. The purpose of this was to ensure that a stable pacing strategy and cadence stabilisation was attained. Due to this applied methodology of the raw data, any surplus braking or cornering that may have occurred during cycling which may have caused significant acceleration spikes would have been reduced. By evaluating this epoch, the authors considered this as a steady baseline measurement. This also accounts for the participants’ anti-clockwise direction around the route in that minimal braking or abrupt cornering would have occurred given the experience of the participants and their familiarity of the route. In a performance context, if one cleat position is more beneficial, it will require less upper body motion and therefore less acceleration magnitudes at a given cadence. In a running context, as less vertical oscillation of the trunk has been proposed to be more advantageous and consequently more efficient, if lesser magnitudes, typically viewed as characteristic sinusoidal curves, are seen in running after cycling in either cleat position, then greater efficiency is inferred. Triaxial trunk acceleration in running was analysed for 60 s at the end of each completed 1 km epoch (i.e., a total of 5 km which represented a complete loop). The root mean square (RMS) values in the three sensing axes were then calculated and used as a measure of the magnitude of trunk acceleration.

### 2.3. Bicycle Cleat Configuration

Two cleat positions were tested. For the MPJ (fore-aft) cleat location used in experiment 1 (day 1), the cleat was placed at a midpoint of the longitudinal difference between the first and fifth metatarsophalangeal joints (MPJ), such that the cleat location lay beneath the third MPJ. The cleat for the posterior (POS) condition (experiment 2, day 2) was located ½ the distance between the control, fore-aft position and the posterior edge of the calcaneus or approximately 1 cm behind the first metatarsal head (POS). To attain the desired POS cleat position, a mechanically engineered aluminium plate was custom designed and manufactured to ensure uniformity between participants and to limit unwanted movement during cycling. The custom-made cleat plate allowed for the aforementioned adjustment (Figure 4).

To accommodate the POS cleat location and modulate foot position, Northwave tri-sonic cycling shoes (Northwave, Via Levada, Pederobba TV, Italy) with Shimano SPD-SL pedals and yellow cleats with a tolerance of approximately 6° flotation and tension were used by all participants with the aluminium plate securely fastened (Figure 5). For both cleat conditions, participants were positioned with the following measurements: 30° knee flexion at point of terminal extension; anterior aspect of patella located vertically over the 3rd MPJ with the cranks in a horizontal position; and relative shoulder joint angle of approximately 90°. All measurements were made with a typical goniometer. To compensate for the effective decrease in leg length with the POS cleat position, the saddle height was positioned 5 mm lower [4].

### 2.4. Bicycle Configuration

Saddle height was measured from the centre of the pedal axle to the saddle top, with the pedal at the most distal end [30] (Figure 6).

Measurements of knee flexion angle were manually taken by the researchers using a typical goniometer with participants in a static, aerodynamic position. Knee flexion was measured with the pedal placed at the bottom dead centre (180°) on the right side of the cyclist at the greater trochanter and lateral femoral condyle. Apart from saddle height, participants did not have bicycle configuration standardised as this would have affected muscle recruitment patterns [31]. Perceptual exertion was described by participants upon completion of each 5 km cycle lap during both experiments using the Borg 6–20 rating of perceived exertion (RPE) scale [32]. The scale is commonly used as a tool for measuring an individual′s effort and exertion, breathlessness and fatigue during physical work. The scale is based on a numerical range from 6–20, where 6 means “no exertion at all” and 20 means “maximal exertion”. The practice of RPE and self-monitoring intensity allows participants to certify that effort is kept within the moderate-intensity range as undue fatigue could cause a confounding effect. All participants had past experience of using perceptual RPE scaling. Time was documented using a Sportline 240 Econosport manual stopwatch (New York, NY, USA).

### 2.5. Statistical Analysis

Statistics were performed using the Analyse-it statistical package (Leeds, UK, version 4.92). The analysis consisted of two parts. Firstly, the general effect of cadence and cleat position on the variable of interest was examined: the intra-individual effects of triaxial trunk acceleration were analysed using a two-way analysis of variance (cadence and cleat position, ANOVA). Ratings of perceived exertion (RPE) were implemented as a covariate. The second and main part of the analysis regarded the effect of cleat position between running: the intra-individual relationships between cleat position (dependent) and triaxial trunk acceleration (independent variable) were analysed using a one-way repeated measures ANOVA. Data normality distribution and sphericity for each accelerometer component were assessed by the Kolmogorov−Smirnov test with a logarithm transform applied to reduce non-uniform data distribution. Threshold values as an effect size (*d*) were used and classified as 0.1–0.3 (small), >0.3–0.5 (moderate), >0.5–0.7 (large), >0.7–0.9 (very large) and >0.9 (extremely large) [33]. Post hoc acceleration analysis was quantified using the Tukey Honest Significant Difference (HSD) test. The significance level was set at *p* < 0.05. For repeatability of measurement, the same sensor and cycle to run protocol was used in experiments 1 and 2. From the accelerometer dataset, cycling cadence changes were annotated during the execution of synchronisation points. This signified the completion of one 5 km cycling lap and cadence condition in order to categorise synchronisation points in the raw data through post hoc analysis. Therefore, in the current dataset, cadence changes were observable due to the corresponding synchronisation points. Longitudinal acceleration was used as an initial indicator of a change to participant trunk acceleration during this analysis.

## 3. Results

The mean acceleration for the POS cleat position in 20 km cycling was 1.93 m/s² compared with 1.45 m/s² in MPJ cycling. When comparing the means of the total group (n = 7), a significant difference (*p* = 0.0001) was found in total acceleration magnitude between cleat positions. There was no significant difference found through RPE between cleat positions (Table 2).

To analyse feasible kinematical variation between MPJ and POS cleat positions during each 5 km lap of cycling and corresponding cadence, data were stratified into x, y, z components. Mean change and RMS can be found in Table 3. A significant difference in greater longitudinal acceleration in the POS cleat position was seen during all laps of cycling with the largest magnitudes for both POS and MPJ observed at the highest cadence of 95–100 rev/min¹. Despite a statistically insignificant result, RPE was higher during the same cadence. In contrast, anteroposterior was lowest throughout all cadences during POS cleat cycling compared to MPJ cycling. Negligible changes to mediolateral motion were observed between both cleat positions. An increase in RMS from the longitudinal axis in POS cycling to MPJ cycling was observed with a simultaneous decrease in anteroposterior acceleration magnitude in POS cycling. Notwithstanding this lack of significance, mean cycling time was quicker during POS cycling at the highest cadence of 95–100 rev/min¹.

Cleat position appeared to influence the longitudinal−anteroposterior relationship, predominantly during the first two laps of cycling. This was tested by comparing the relationship during post-hoc analysis. Despite a very large effect size (*d* = 0.82), ANOVA showed that statistical significance was not detected between the longitudinal−anteroposterior magnitudes between MPJ and POS cleat cycling (*p* = 0.052) with a mean difference of 0.89 m/s² (±1.2) (Figure 7).

### Running after Cycling

The results for running after cycling in an MPJ cleat position and a POS cleat position are presented in Table 4. Mean trunk acceleration magnitude in running from a POS cleat position was significantly lower compared to the MPJ position, with a large effect size. Similarly, effect size was extremely large and with a significant statistical outcome in RPE.

Figure 8 shows triaxial acceleration magnitudes in 5 km running after both MPJ and POS cycling, with running after POS cycling being of a lower acceleration magnitude.

Between-running variables (x, y, z) for each 1 km of running were compared between conditions (running from a POS and MPJ cleat position). Although mean longitudinal acceleration was of a lesser magnitude during running after a POS cleat compared with running post-MPJ cycling, minimal differences were seen in both mediolateral and anteroposterior motion aside from the second km of running. Interestingly, the average 5 km run time was significantly faster in running after POS cycling (Table 5).

For a better understanding of trunk kinematics during running, Figure 9 shows the variability of trunk magnitude in the two experimental tests. The increase in mediolateral acceleration in running from POS cycling during 5 km of running is statistically significant.

## 4. Discussion

The purpose of this sensor-based approach study was to characterise the change in trunk acceleration magnitude in triathletes instructed to cycle with an MPJ cleat position and POS cleat position. The secondary purpose was to determine if a POS cleat position is more beneficial due to lower magnitudes of trunk acceleration when running from cycling. This original technique by means of sensor technology could be applied to provide race-standard feedback on accelerations of the trunk in triathlon cycling and running whilst contributing to prior research concerning the benefits of a more posteriorly-orientated bicycle cleat. The capability of a non-invasive sensor to distinguish variations in acceleration magnitude of the trunk in different cleat positions in cycling and sinusoidal curvatures in running was matched with perceptual RPE scaling, with a significant reduction of perceived exertion reported when running from a POS cycle cleat position.

The results of this study highlight two important changes in temporal trunk kinematics in cycling and running, which, in part, may explain the improvements in running from a POS cleat position previously reported. Despite the reduced trunk acceleration in running after cycling in a POS position, POS cycling significantly increased the magnitude of trunk acceleration, albeit with a moderate effect size (Table 2). These kinematical deviations of trunk acceleration were associated without causing significant alteration to RPE. Furthermore, the variations of trunk acceleration occurred in the longitudinal direction with nominal increases seen in mediolateral motion. Despite prior studies confirming a beneficial effect of antero-posterior (i.e., MPJ-POS) cleat placement on physiological variables in running [4,14,19], few studies have assessed the temporal kinematics of the trunk and the corresponding impact of cycling and running. This makes direct comparisons with the results presented in this study difficult. McDaniel et al. [34] demonstrated that an aft cleat position induced a higher activity of prominent muscles tibialis anterior (TA), vastus lateralis and gluteus maximus. Moreover, Litzenberger et al. [21] found a lower activity of the TA during the first part of the downstroke (between 0° to 40° of the pedalling cycle) when the cleat centre was aligned with the first metatarsal head compared to a mid-foot cleat position. While the higher values of longitudinal trunk acceleration in POS cycling in the current study cannot be explained by muscular activity, theoretically the trunk could compensate if the lower limbs changed position and consequently changed the length−tension relationship. Research is needed to test this proposition, however.

It is interesting to note that moving the cleat posteriorly during cycling was associated with lower magnitudes of anteroposterior acceleration (Figure 7). This was apparent despite an overall increase in longitudinal motion in POS cycling. The reasons for greater longitudinal yet reduced anteroposterior acceleration are not wholly understood, but it can be speculated that such a change may produce alterations in general cycling efficiency. In this sense, anteroposterior motion represents the global (gross) movement of the triathlete during cycling. An alteration in upper body position is related with changes in activation of lower limb muscles [7] whilst changes in cadence can optimise or deteriorate gross and physiological efficiency [35]. This implies that when cycling at different cadence ranges, triathletes self-select trunk position in order to maintain cadence. However, performing high or low cadence ranges may lead to suboptimal technique and poor trunk motion, resulting in insufficient muscle recruitment and force production, and consequently, a disadvantaged performance. More complex IMUs combined with power and joint torque calculations may give improved estimates.

Within the range of trunk acceleration magnitudes used by this group of triathletes, there is a clear relationship between cadence and RPE, and even more so at cadences of 95–100 rev/min¹ (Table 3). Evidence of this can been seen in prior investigations concerning cadence, pedalling technique and gross efficiency in cycling [36]. Still, intra-variability relative to cadence is likely dependent on additional variables such as gross efficiency, crank inertial load and physiological capability.

Previous studies have performed cycling tests with a 5-mm posterior (PCP) and a 5-mm anterior (ACP) first metatarsal cleat position [16,21,36]. Notably, Millour et al. [14] evaluated biomechanical and physiological variables of a simulated a sprint triathlon and detected no significant change in cycling power output. Although tests included 30 s sprints performed at maximal aerobic power, the lack of a significant difference between VO_2_ and power output in both cleat positions suggests that pedal force effectiveness is not compromised when cycling. This may be explained in a similar way when considering the current study. The lack of statistical significance and small effect size seen in RPE in the present study irrespective of cleat position could indicate that pedal force effectiveness is not compromised. If, theoretically, RPE increased, it could designate a reduction in force, power or cadence due to increased exertion. As this was outside the scope of the current study, further research is needed to verify this statement.

### Running after Cycling

The magnitude of longitudinal trunk acceleration in the 5 km self-selected pace run after transitioning from POS cycling significantly decreased compared to running after MPJ cycling (Table 4). Moreover, a large effect size was seen between conditions, which is indicative of significant changes to temporal trunk acceleration magnitudes. Taken together with the significant decrease in RPE when running from POS cycling and a quicker 5 km run time, it would appear that POS cycling could be advantageous when running is required immediately afterwards. Of the post hoc analysis, the range of acceleration magnitudes across each 1 km epoch revealed an association between lower magnitudes of longitudinal acceleration in POS running (Table 5, Figure 8 and Figure 9). It may be tempting to conclude that the reduced longitudinal acceleration and RPE in running was caused by cycling in a POS cleat, or in other words, that changing the cleat setback resulted in less perceived exertion alongside less oscillation. However, this is unlikely because of the increases to longitudinal acceleration seen in POS cycling. The decreased biomechanical cost due to a smaller amount of gross motion (i.e., anteroposterior movement) during POS cleat cycling is a more likely explanation. There are no studies, however, that have investigated the amount or relationship of longitudinal−anteroposterior trunk acceleration when moving from cycle to run. However, various biomechanical assessments have shown the performance benefits of using a more POS-orientated cleat in triathlon. Paton and Jardine [4] demonstrated the benefit of using a POS cleat for a subsequent 5.5 km treadmill run. Results from this study are supported by previous findings that were performed in laboratory conditions [3,12] or in outdoor conditions with cycling and running distances that are similar to those in the current investigation [11].

It seems plausible that if the longitudinal acceleration of the trunk is reduced, or if movements do not diverge from the running direction, an improved running time may result. In running, there are energy transfers occurring in each cycle between kinetic and potential energy, both in the form of the body COM height and stretched or compressed elastic components [37]. The body COM is acknowledged to be a key determinant of the characteristic spring−mass behaviour in running. A runner′s leg stiffness during the stance phase controls the vertical motion of the COM [38]. As leg stiffness increases, vertical motion of the COM decreases, stride frequency increases and foot−ground contact time decreases [39]. In this sense, the magnitude and direction of ground reaction force are determined by the acceleration and position of a runner′s COM [40]. From a practical standing, the changes seen in the current study are significant in that running with a POS cleat reduced the magnitude of trunk acceleration. This is quite a noteworthy finding as it suggests that the upper body of the individual triathlete is affected by lower body orientation (i.e., cleat position). Thus, by reducing the magnitude from the upper body essentially ‘bouncing’, a runner may decrease the peak vertical GRF and reduce the energetic cost of running [41]. However, the actual amount of energy that is dissipated during running (and hence must be compensated by work of the muscles) is not entirely known. Nevertheless, if we assume that the amount of energy dissipated correlates positively with the actual fluctuation of potential energy, it makes sense to focus on trunk displacement and acceleration magnitudes as relevant for running efficiency.

The changes between mediolateral motion may appear marginal, although an exception was detected during the final km of running with a significant increase in mediolateral movement in POS running. Thorstensson et al. [42] showed that movements in the mediolateral directions are smaller than in the longitudinal direction and probably have less influence on running efficiency. The fact that there is a difference implies that the magnitude of acceleration relative to mediolateral motion moves with respect to the anteroposterior reference frame. Recently, a comprehensive observational study related kinematic variables to runners′ best times and the metabolic cost of running. It was found that there is a large amount of variation in the magnitude of vertical motion between different runners and that the differences in vertical motion of the pelvis during ground contact is strongly correlated with both performance measures [43]. Despite the evidence suggesting an association between longitudinal trunk acceleration and running economy, to the best of the authors’ knowledge, there are no studies that have quantified changes to trunk motion in different cycling positions and how those changes relate to running performance in triathlon. However, results in the current study suggest that the magnitude of trunk acceleration, particularly longitudinal acceleration, appears to be affected by prior cycling. Whilst the current study measured two cleat positions and the influence on running, others have suggested that prior cycling does not cause changes in variables such as ground contact time, and notably, vertical oscillation. Future research using accelerometry combined with a force platform and spatiotemporal measures could provide additional clarity. Along this line, conjecture exists as to if the bike−run transition affects physiological parameters more than biomechanical parameters [3,7]. Such findings (i.e., physiological−biomechanical) were not reproduced in the present study, which leaves this proposed explanation open for debate. It is important to consider that this study was designed to measure the immediate effect of altering cleat position during cycling and the impact on running using a portable triaxial accelerometer. The use of commercially available wearable devices to measure cycling and running kinematics is on the rise. These devices provide clinicians with dependable and accurate methods to monitor data in real-time and in the athlete’s usual training and competitive environment. The results of this study show that changes in bicycle cleat position measured with a wearable device can demonstrate variations in lower limb loading. Sports clinicians may be able to utilise these devices to provide training feedback outside of laboratory or clinical settings and to monitor athletes’ ability to alter temporal trunk acceleration magnitudes and compliance with the prescribed interventions.

The current study is not without limitations, including small sample size and lack of an elite population. Future studies should look to assess full kinematic, kinetic and loading rate parameters associated with changes in spatiotemporal measures in both cycling and running when cycle geometry is altered. Future studies should also look to see long-term outcomes for changes in metabolic demand. Furthermore, the training age, experience, physiological capability and the possibility of learning effects may have influenced pedalling dynamics and therefore running performance. Despite this, the current results show the benefit of using wearable technology to accomplish analysis that allows monitoring for prolonged durations in a natural training setting.

## 5. Conclusions

In the case of trunk acceleration magnitude in two different bicycle cleat positions, the evaluation of data outputs from a triaxial accelerometer in triathlete cycling could be practically relevant. Despite a greater mean magnitude of longitudinal trunk acceleration during posterior cleat cycling compared to the characteristic metatarsophalangeal/fore-aft position, this did not impact running immediately after cycling. Reduced magnitudes of longitudinal acceleration were detected when running after posterior cleat cycling with the inference that less magnitudes of vertical oscillation improved subsequent running time. An unobtrusive wireless tri-axial accelerometer with the ability to continuously measure trunk accelerations during an outdoor, varied yet augmented cadence cycle could be relevant for postural considerations when exploring cleat positions and running immediately after cycling in a triathlon. Whereas the difference to RPE was nominal during cycling, a significant decrease was seen in running after cycling during posterior cleat placement. Further research is warranted to assess changes in performance settings.

## Figures and Tables

**Figure 1 sensors-21-05899-f001:**
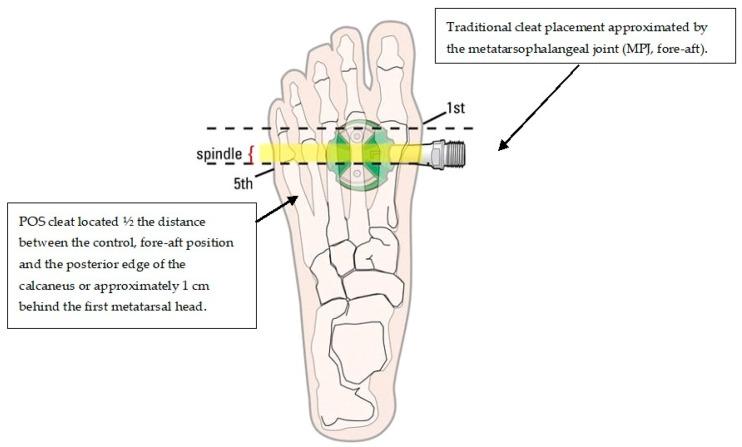
Representation of two different cleat locations (not to scale). Retrieved from bikefit.com (accessed on 31 August 2021).

**Figure 2 sensors-21-05899-f002:**
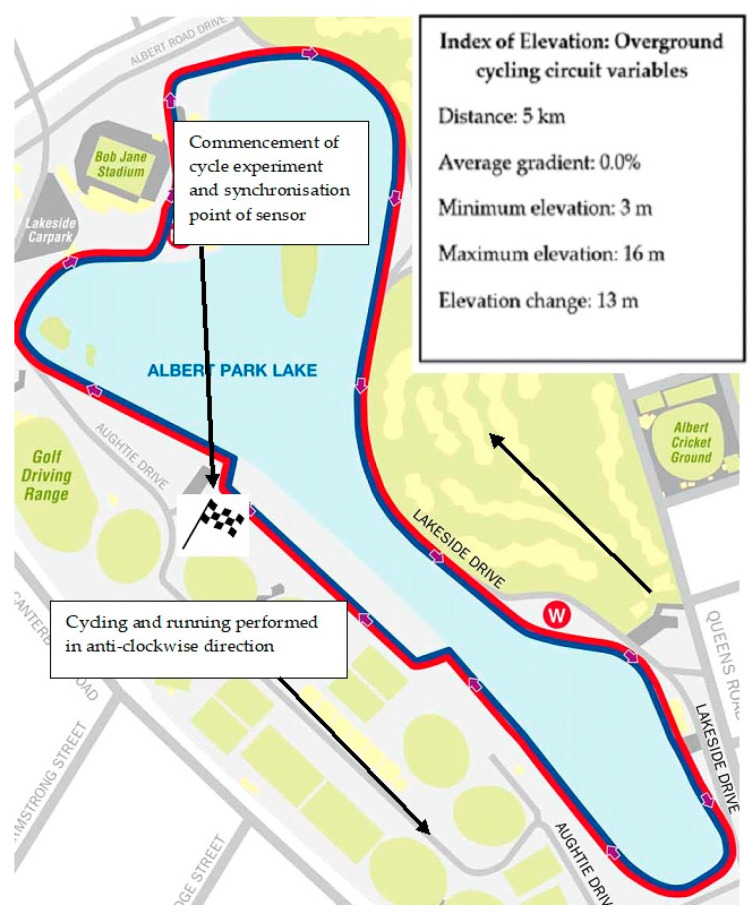
Map view and location of both experiments. The index of elevation contains the course variables experienced by the participants during both experiments. The average gradient across the 5 km circuit was 0%.

**Figure 3 sensors-21-05899-f003:**
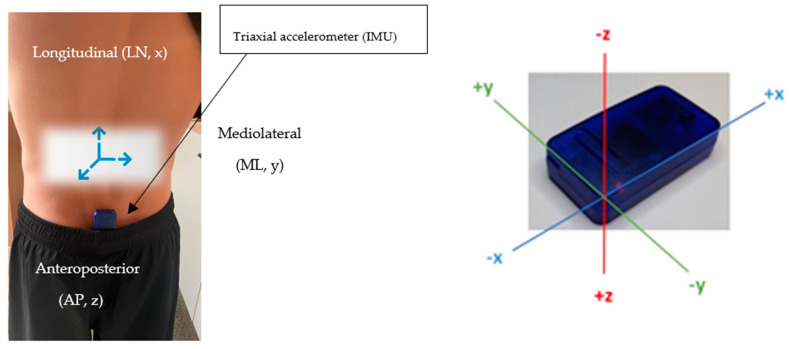
Representation of orthogonal axes orientation and sensor used in study (**left**). Actual sensor used in study (**right**).

**Figure 4 sensors-21-05899-f004:**
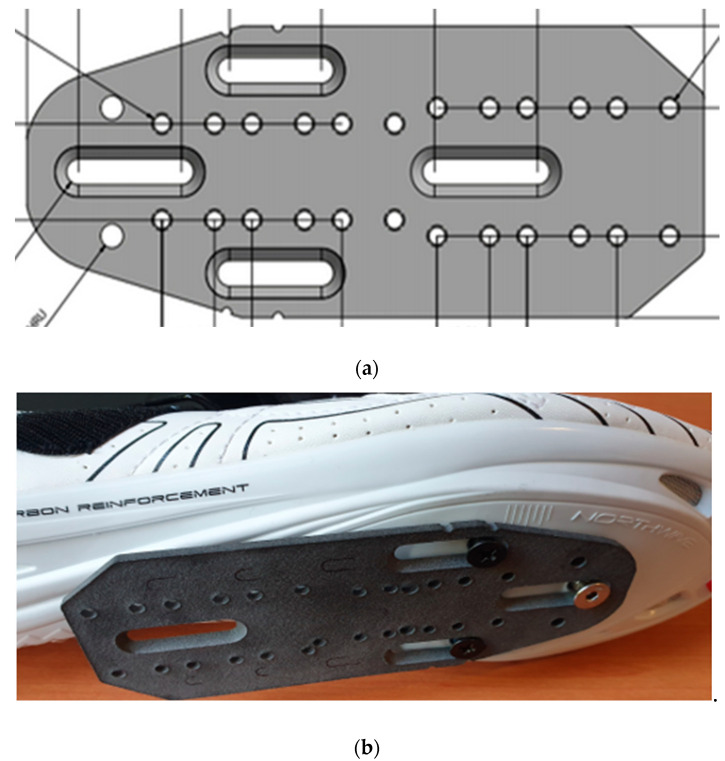
(**a**) Engineering CAD illustration including specifications and dimensions of aluminium bicycle cleat plate; (**b**) Prototype of cleat plate attached to cycling shoe.

**Figure 5 sensors-21-05899-f005:**
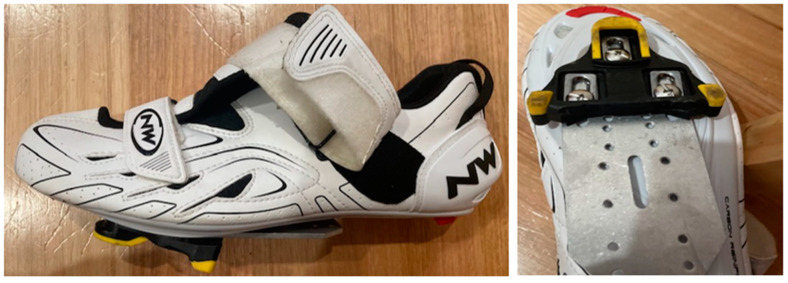
Northwave tri-sonic cycling shoe used in study with cleat plate securely fastened. Cleat position shown is POS with yellow Shimano SPD-SL cleat securely fastened to the cleat plate.

**Figure 6 sensors-21-05899-f006:**
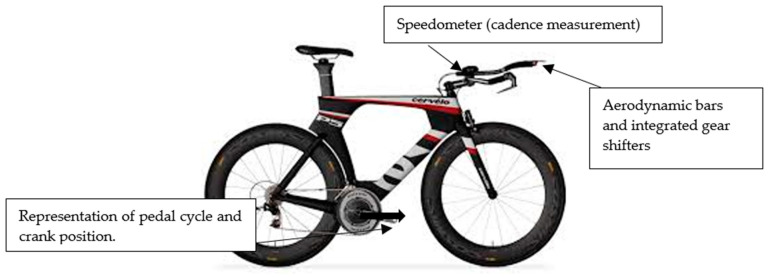
Example of triathlon bicycle used in study. Model shown is the Cervelo P5 which was used by one participant.

**Figure 7 sensors-21-05899-f007:**
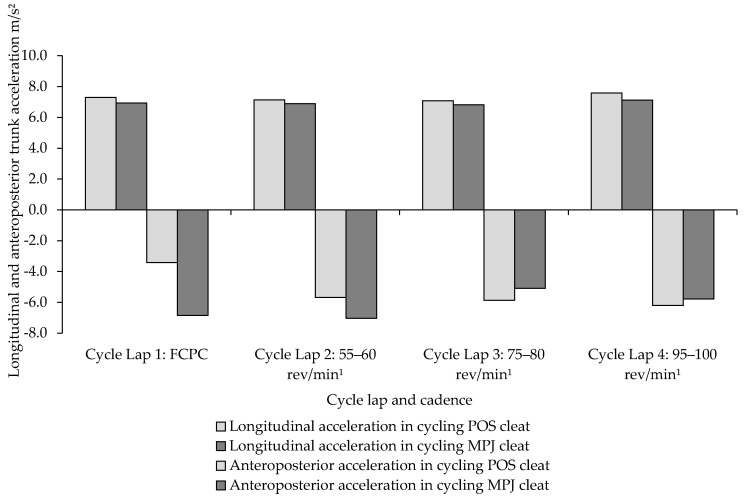
Comparison of the longitudinal−anteroposterior trunk acceleration relationship between MPJ and POS cleat positions in 20 km of cycling.

**Figure 8 sensors-21-05899-f008:**
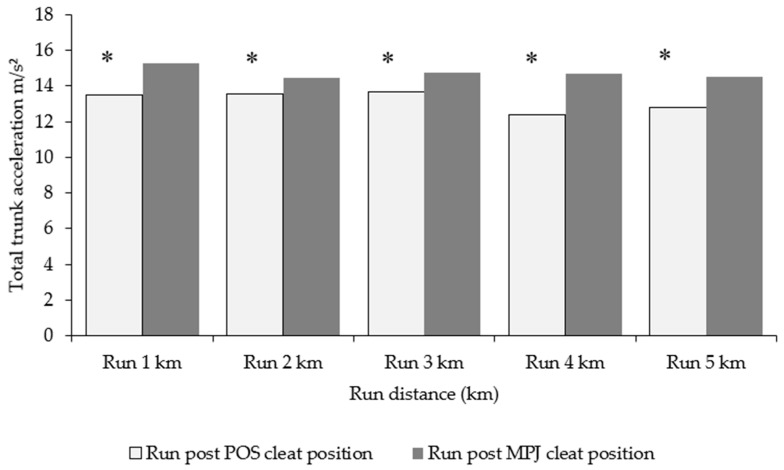
Total triaxial accelerations of the trunk during each 1 km epoch of running after cycling in MPJ and POS cleat positions. Acceleration presented in m/s². * Significant at *p* < 0.05.

**Figure 9 sensors-21-05899-f009:**
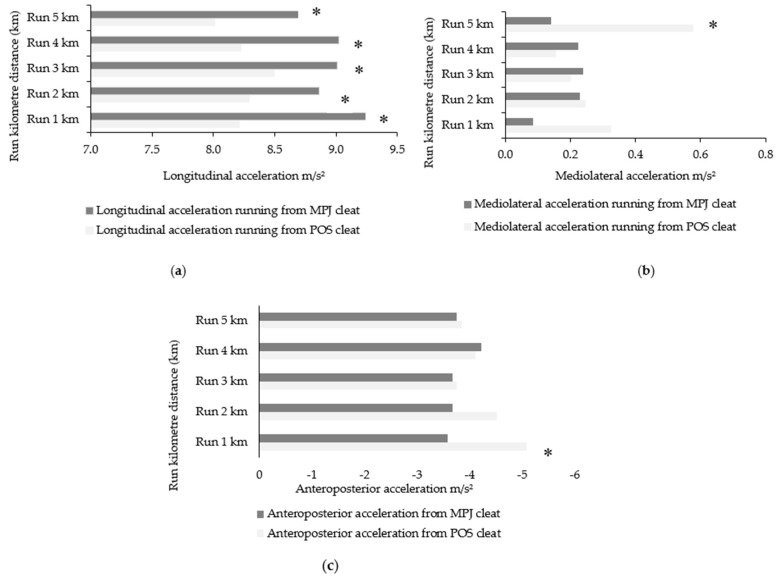
(**a**): Longitudinal acceleration magnitudes per kilometre in running after POS and MPJ cleat positions; (**b**) Mediolateral acceleration magnitudes per kilometre in running after POS and MPJ cleat positions; (**c**) Anteroposterior acceleration magnitudes per kilometre in running after POS and MPJ cleat positions. POS = posterior cleat. MPJ = metatarsophalangeal joint cleat position. * Significant at *p* < 0.5.

**Table 1 sensors-21-05899-t001:** Cadence (in rev/min^1^) protocol performed for experiment 1 and experiment 2. Cadence was fixed (i.e., not randomized) for each lap.

Duration (Epoch)	Lap 1	Lap 2	Lap 3	Lap 4
	5 km	5 km	5 km	5 km
Cadence	Freely Chosen Pedaling Cadence (FCPC)	55–60 rev/min^1^	75–80 rev/min^1^	95–100 rev/min^1^

**Table 2 sensors-21-05899-t002:** Magnitude of mean trunk acceleration (in m/s^2^) in two cleat positions in 20 km cycling. RPE = ratings of perceived exertion.

.	MPJ Cleat Position	POS CleatPosition	Mean Difference ± SD	*p* Value	Threshold Effect Size and Magnitude Inference
Mean acceleration m/s²	1.45	3.38	1.93 (±2.15)	0.0001 *	>0.7 (large)
RPE	10.2	10.3	0.1 (±2.2)	>0.05	0.1 (small)

* Significant at *p* < 0.5.

**Table 3 sensors-21-05899-t003:** Descriptive data (means ± SD) from individual 5 km laps of the overground cycle in two cleat positions (in m/s^2^). ^1^ RMS = root mean square. ^2^ RPE = ratings of perceived exertion. ^†^ Mean Diff = difference between MPJ and posterior (POS) cleat positions. Lap time reported in minutes.

	**MPJ (Fore-Aft) Cleat Position**					**POS Cleat Position**				
Lap 1 (FCPC)	x	y	z	RPE ²	Lap Time	x	y	z	RPE ²	Lap Time
^†^ Mean acc m/s²	6.95 * (±1.2)	−0.08 (±0.4)	−6.84 * (±3.3)	8.5 (±0.5)	10.46 (±0.37)	7.29 * (±1.8)	0.48 (±0.6)	−3.42 * (±4.7)	8.8 (±0.6)	10.48 (±.038)
Lap 2 (55–60 rev/min^1^)										
^†^ Mean acc m/s²	6.89 * (±1.0)	−0.03 (±0.4)	−7.03 * (±2.8)	9.8 (±0.6)	9.42 (±0.50)	7.14 * (±1.6)	0.19 (±0.4)	−5.68 * (±2.3)	10.1 (±0.5)	9.50 (±0.51)
Lap 3 (75–80 rev/min^1^)										
^†^ Mean acc m/s²	6.81 * (±1.1)	−0.08 (±0.4)	−6.20 * (±1.2)	10.8 (±1.0)	9.12 (0.61)	7.08 * (±1.4)	0.12 (±0.4)	−5.86 (±2.2)	10.8 (±1.2)	9.18 (0.52)
Lap 4 (95–100 rev/min^1^)										
^†^ Mean acc m/s²	7.11 * (±1.2)	−0.05 (±0.5)	−5.78 * (±1.7)	12.1 (0.8)	9.02 (0.49)	7.59 * (±1.5)	0.28 (±0.4)	−5.09 (±2.4)	11.5 (±1.2)	8.59 (0.70)
RMS ¹	8.51	1.05	4.01			9.01	1.01	3.72		

* Significant at *p* < 0.5.

**Table 4 sensors-21-05899-t004:** Descriptive data (means ± SD) from individual 5 km laps of overground running after cycling in two cleat positions (in m/s^2^). ^1^ RPE = ratings of perceived exertion. ^†^ Diff = difference between MPJ and POS cleat positions.

	Run Post MPJ Cleat Position	Run Post POS Cleat Position	^†^ Mean Difference ± SD	*p* Value	Threshold Effect Size and Magnitude Inference
Mean acceleration m/s²	8.50 (±2.7)	6.21 (±2.8)	2.3 (±1.1)	0.0001 *	0.9 (large)
RPE ¹	12.71 (±0.48)	11.85 (±0.37)	0.1 (±1.7)	0.0167 *	>0.1 (extremely large)

* Significant at *p* < 0.5.

**Table 5 sensors-21-05899-t005:** Descriptive data (means ± SD) from individual 5 km run post cycling in two cleat positions (in m/s^2^). ^1^ RPE = ratings of perceived exertion. ² Time in minutes = mean time for completion of 5 km run. ^3^ RMS = root mean square.

	Run Post MPJ Cleat Position			Run Post POS Cleat Position		
1 km	x	y	z	x	y	z
Mean acc m/s²	9.3 * (±1.6)	0.01 (±0.6)	−3.5 (±2.3) *	8.2 * (±1.6)	0.3 (±0.6)	−5.1 (±4.7) *
2 km						
Mean acc m/s²	8.8 * (±1.3)	0.2 (±0.5)	−3.6 (±2.2)	8.2 (±1.6)	0.2 (±0.4)	−4.5 (±2.3)
3 km						
Mean acc m/s²	9.0 * (±0.7)	0.2 (±0.6)	−3.7 (±1.3)	8.4 * (±1.4)	0.2 (±0.4)	−3.7 (±2.1)
4 km						
Mean acc m/s²	9.1 * (±1.6)	0.2 (±0.7)	−4.2 (±1.8)	8.2 * (±1.5)	0.1 (±0.4)	−4.1 (±2.4)
5 km						
Mean acc m/s²	8.7 * (±1.7)	0.1 (±1.4) *	−3.7 (1.4)	8.0 * (±1.7)	0.5 (±1.4) *	−3.8 (±1.4)
RMS ^3^	10.6	4.3	2.8	10.2	1.1	2.6
RPE ¹	12.7 (±3.8)			11.8 (±4.8)		
Time (minutes) ²	22.6 (±1.9)			22.1 (±2.0)		

* Significant at *p* < 0.5.

## Data Availability

The data presented in this study are possibly available on request. This will be subject to institutional ethical approval for release by the researchers.

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
