# Peer review of "The Effect of Cleat Position on Running Using Acceleration-Derived Data in the Context of Triathlon"

_sensors, 2021, doi:10.3390/s21175899_

Round 1

Reviewer 1 Report

This is a well written and interesting paper which should be accepted for publication with only minor editorial correction.

There was concern regarding the sample size. However the authors have have addressed this concern by the acknowledgement of this limitation within the body of the paper.

There are minor editorial errors such as the number '84' in Figure 1 and the units being separated the abbreviation such as '12' and 'mm' at line 206 and 'Table' and '4' at line 502. The authors need to address all occurrences of these prior to publication.
Define rating of perceived exertion (RPE) at first occurrence in text and type in full with Abstract.
For Figure 3, change ‘Triaxial accelerometer' to IMU or add IMU within brackets 'Triaxial accelerometer (IMU)’.

Author Response

Point: This is a well written and interesting paper which should be accepted for publication with only minor editorial correction. There, was concern regarding the sample size. However, the authors have addressed this concern by the acknowledgement of this limitation within the body of the paper.

Response: The authors thank the reviewer for the constructive comments and suggestions. Detailed responses are provided below.

Point 1: There are minor editorial errors such as the number '84' in Figure 1 and the units being separated the abbreviation such as '12' and 'mm' at line 206 and 'Table' and '4' at line 502. The authors need to address all occurrences of these prior to publication.

Response 1:  The authors thank the reviewer. The authors confirm that the above changes have been made.

Point 2: Define rating of perceived exertion (RPE) at first occurrence in text and type in full with Abstract.

Response 2: The authors have revised the Abstract to ensure RPE is defined, in full, at the first occurrence. The authors thank the reviewer.

Point 3: For Figure 3, change ‘Triaxial accelerometer' to IMU or add IMU within brackets 'Triaxial accelerometer (IMU)’.

Response 3: The authors confirm that this change has been made. The reviewer is advised that Figure 3 has been amended in order to improve clarity.

Reviewer 2 Report

Very good methodology and presentation. I recommend, however, repeating this study with a l9arger number of (well-)trained subjects.

Author Response

Reviewer 2

Point 1: Very good methodology and presentation. I recommend, however, repeating this study with a larger number of (well-)trained subjects.

Response 1: The authors thank the reviewer for the constructive comments. The small sample size in this study was limited due to COVID restrictions during the time of the research design and participant recruitment process, however, the authors agree that it does set the foundation for a larger and more homogenous cohort.

Reviewer 3 Report

I think defining RPE in the abstract makes sense rather than waiting until line 319 since many readers will only read the abstract.

I'm not sure reporting p-values to the number of significant figures (i.e. p=0.0369 on line 25) is appropriate for n=7 when p=0.04 would suffice and more accurately reflects the strength of the data.

The closing lines of the abstract seem to dilute the impact of your findings. Of course accelerometers can measure body motion, that has been shown a million times. What is unique about your findings and what do they imply in terms of changes in triathlon practice and suggest for future experimental work?

Line 44: efficiency should be efficiently

Line 55 suggests that reducing trunk COM motion in cycling would be advantageous, while your results in the abstract indicate an increase in COM motion while at the same time reducing RPE. Is this discrepancy or surprise finding discussed elsewhere?

Figure 1 is crowded by the text boxes and makes evaluating the location of the anatomical features under discussion difficult. Please revise, making the entire foot visible.

Similarly, Figure 2 seems unnecessarily crowded by the text boxes which overlap portions of the image. Please adjust.

It is unclear to me why the aero position enabled by tri-bars limits the need for braking (line 197) beyond the need for breaking posture to reach the brake levers. It would seem that riding speed and sharpness of turns or presence of traffic or other obstacles would impact braking rather than riding posture. A rider in traditional road posture on the same course would not brake unless such obstacles present or turns were needed as braking is counterproductive to racing regardless of posture.

The text of Figure 3 again could be improved in terms of placement and contrast to ease reader comprehension. Also, the photo of the rider's back is small and low quality. Can a better picture or schematic cartoon be used to clarify? The use of actual human back doesn't add much, and a multi-view schematic would more clearly explain the setup.

Line 248: Likened? Not sure what this means. Collected, perhaps?

Line 270: Custom, not customed.

Figure 4: The large number of dimensions shown on the CAD drawing are not helpful and serve to clutter the figure. Please highlight the features of import only. Also, the text box "threading to permit..." points to a slot which clearly cannot be threaded. Please correct.

Line 300: Saddle height is a parameter that has a strong impact on cycling kinematics. Applying a 5mm change in height uniformly across all subjects is appropriate? By what analysis? The value would seem to be dependent on the stature and proportions of each rider. How can you remove the impact of saddle height changes from the data? Have experiment 3 with POS cleat at original saddle and experiment 4 with MPJ cleat at lower saddle?

Figure 5 shows the cleat plate on the shoe. Was the increase in effective leg length resultant from the added height of the cleat plate accounted for in setting saddle height?

Figure 6 again has text boxes obscuring the image and the text of the rightmost box appears to be truncated as "...integrate gear shift levers" would seem to be the appropriate text.

Line 343: The longitudinal acceleration did not detect the cleat position. The cleat position was found to impact the longitudinal acceleration. Were any significant differences in acceleration observed in the other axes? Ahh, I see this in lines 358. What could the reduction of anteroposterior mean?

Line 344: Main peak? This seems to be referring to data or figures not available to the reader and is therefor superfluous. Perhaps a figure of an acceleration waveform is valuable to explain the methodology.

Line 346: "longitudinal acceleration for the" is needed between mean and POS.

Table 3 seems to contradict the data in Table 2, where the mean accelerations are much less than reported in Table 3. Also, Table 3 seems to show that the difference in anteroposterior acceleration (z-axis) is larger as a percentage than longitudinal (x-axis) between the POS and MDJ positions. Please clarify and explain.

The labels in Figure 7 are nonsense. NTL is used instead of MDJ and 3 of the 4 data sets are labeled POS. Correct please.

Again Figure 8 uses the otherwise unmentioned NTL nomenclature. Correct.

Figure 9 uses "smooth" lines between data points. This implies a model of some sort and should be avoided. The data should not have lines between them, as no data is available nor is a prediction based on a model available. Rather, each data point at the check points should include the error bars from the n=7 subjects. Without error bars Figure 9 is useless. Line 410 references variability, and variability can only be assessed with error bars which represent the data variability.

Additionally, Line 411 refers to the slope of the difference in mediolateral acceleration. You have no information as to the slope. No data is presented between 4-5km, and while the smoothed line has a slope, your data does not.

Line 446: What is disproportionate? How do your results apply to an individual athlete?

Line 455: Your data showed an increase in longitudinal acceleration but also a larger reduction (on a percentage basis) of anteroposterior acceleration when cycling in the POS as compared to the MDJ cleat position. The relative impact of the two differences seem to be ignored in your paper. Please expand.

Why not take the next step and calculate the multi-axis COM motion as an energy expenditure and compare the POS and MDJ positions? Motion of the COM is real expenditure. With continuous accelerometer readings this should be simple.

Finally discussion of anteroposterior is found in 470. This review was conducted in stream of consciousness, as read, so please bear with me.

Line 538: Why not continue this train of thought? Can the acceleration data not be extended as a proxy for energy expended based on assumptions of other factors being equal from session 1 to 2 on the kinematics of the athletes? If overall COM motion is reduced with one or the other cleat position, then one or the other is more efficient.

Line 577: Please clarify that this applies to only one accelerometer axis, not COM acceleration as a whole.

Author Response

The authors thank the reviewer for the constructive and helpful comments. The authors have made every effort to address the reviewer’s comments and hope that the responses and improvements meet expectations.

Point 1: think defining RPE in the abstract makes sense rather than waiting until line 319 since many readers will only read the abstract.

Response 1: The authors have amended the abstract to reflect these changes. The authors thank the reviewer.

Point 2: I'm not sure reporting p-values to the number of significant figures (i.e. p=0.0369 on line 25) is appropriate for n=7 when p=0.04 would suffice and more accurately reflects the strength of the data.

Response 2: The authors agree with the reviewer and have consequently amended the p-value.

Point 3: The closing lines of the abstract seem to dilute the impact of your findings. Of course, accelerometers can measure body motion, that has been shown a million times. What is unique about your findings and what do they imply in terms of changes in triathlon practice and suggest for future experimental work?

Response 3: The authors thank the reviewer for the constructive comments. The authors have revised the final two paragraphs from the abstract. It now states:

Page 1 Abstract:

Practically, the magnitude of trunk acceleration during cycling in a posterior cleat position as well as running after posterior cleat cycling differed from that when cycling in the fore-aft position followed by running. Therefore, the notion that running varies after cycling is not merely an individual athlete’s perception, but a valid observation that can be modified when cleat position is altered. Training specifically with a posterior cleat in cycling might improve running performance when trunk accelerations are reduced.

Point 4: Line 44: efficiency should be efficiently.

Response 4: The authors thank the reviewer for highlighting this error, which has now been amended accordingly.

Point 5: Line 55 suggests that reducing trunk COM motion in cycling would be advantageous, while your results in the abstract indicate an increase in COM motion while at the same time reducing RPE. Is this discrepancy or surprise finding discussed elsewhere?

Response 5: Once again the authors thank the reviewer for the considered response. An increase in longitudinal acceleration was not necessarily a surprise in cycling with a POS cleat compared with the MPJ position. Whilst the effect size was moderate and RPE was small (and not significant by statistical standards), a relative change in trunk acceleration magnitude would be expected to compensate for the effective decrease in leg length with the POS cleat position. In this instance the saddle height was positioned 5 mm lower in line with Paton & Jardine (2012) (Figure 5). The suggestion that a reduction in trunk acceleration magnitude when cycling would be advantageous was speculative. However, as it is known that trunk position (e.g., angles, configuration) effects both physiological and lower limb performance during cycling, the rationale that higher magnitudes of trunk acceleration essentially constitute ‘unwanted or wasted motion’ is built on prior research that upper body movement can influence and alter contributions from the knee and hip [Bini et al., 2019. [Muscle force adaptation to changes in upper body position during seated sprint cycling. Journal of Sports Sciences, 37(19); Grant et al., 2014. The Effect of Prior Upper Body Exercise on Subsequent Wingate Performance, BioMed Research International, https://doi.org/10.1155/2014/329328; Fang et al., 2014. Effects of Cycling Workload and Cadence on Frontal Plane Knee Load. Master's Thesis, University of Tennessee].

Point 6: Figure 1 is crowded by the text boxes and makes evaluating the location of the anatomical features under discussion difficult. Please revise, making the entire foot visible.

Response 6: Figure 1 has been revised based on the reviewer’s recommendation.

Point 7: Similarly, Figure 2 seems unnecessarily crowded by the text boxes which overlap portions of the image. Please adjust.

Response 7: Figure 2 has been revised based on the reviewer’s recommendation.

Point 8: It is unclear to me why the aero position enabled by tri-bars limits the need for braking (line 197) beyond the need for breaking posture to reach the brake levers. It would seem that riding speed and sharpness of turns or presence of traffic or other obstacles would impact braking rather than riding posture. A rider in traditional road posture on the same course would not brake unless such obstacles present or turns were needed as braking is counterproductive to racing regardless of posture.

Response 8:  The authors express thanks to the reviewer for this response. The bicycles used in the study were classified as triathlon, or time-trial bicycles, that comprised steeper seat tube angles compared to road bicycles This has an effect on the triathlete’s position, generally bringing the rider further forward over the bottom bracket. This, in theory, allows the triathlete to reduce some of the cycling impact in better preparation for the run phase. Additionally, on a triathlon bike, the steering column is usually steeper, giving a stiffer feel to the front of the bicycle and thus making it more responsive round corners. This also has the effect, together with an often-tighter rear triangle, of creating a shorter wheelbase than a traditional road bike. Apart from standard group-set parts such as brake callipers and chain sets, which are also common on a road bike, other components on a triathlon bicycle include a set of fixed aerobars, lightweight brake levers and bar-end gear shifters. These allow the triathlete to essentially cycle with one hand and arm placement and, therefore, one riding position. In this instance, the brake levers are located posteriorly to the aerobars and requires a deliberate change in hand and therefore trunk position to either reach for the front and/or rear brake. This is in contrast to a road bicycle whereby the brake levers are located directly in front of the drop bars, therefore staying relatively close to the intended cycling position. Despite this, the authors have revised the original paragraph to make this definition clearer and avoid misinterpreting the situation. The authors hope that the reviewer is satisfied with the revision

Page 5, Line 247

The bicycles used in the study were classified as triathlon, or time-trial bicycles, that comprised steeper seat tube angles compared to road bicycles. This has an effect on the triathlete’s position, generally bringing the rider further forward over the bottom bracket. Apart from standard group-set parts such as brake callipers and chain sets, which are also common on road bicycles, the fixed aerobars, lightweight brake levers and bar-end gear shifters found on triathlon bicycles allow the triathlete to essentially cycle with one hand and arm placement and, therefore, one riding position. In this instance, the brake levers are located posteriorly to the aerobars and requires a deliberate change in hand and therefore trunk position to either reach for the front and/or rear brake. This is in contrast to a road bicycle whereby the brake levers are located directly in front of the drop bars, therefore staying relatively close to the intended cycling position. In this sense, the aerodynamic position generally places more reliance on using the integrated gearing shifters located at the end of the aerodynamic bars, which differs from that used by road cyclists. Therefore, triathletes are likely to “shift up or down” to a lesser gearing ratio to reach a self-selected and efficient cadence to preserve performance.

Point 9: The text of Figure 3 again could be improved in terms of placement and contrast to ease reader comprehension. Also, the photo of the rider's back is small and low quality. Can a better picture or schematic cartoon be used to clarify? The use of actual human back doesn't add much, and a multi-view schematic would more clearly explain the setup.

Response 9: Figure 3 has been revised based on the reviewer’s recommendation.

Point 10: Line 248: Likened? Not sure what this means. Collected, perhaps?

Response 10:  Likened has been replaced with collected in order to remove ambiguity. The authors thank the reviewer.

Point 11: Line 270: Custom, not customed.

Response 11:  The authors confirm that this has been amended.

Point 12: Figure 4: The large number of dimensions shown on the CAD drawing are not helpful and serve to clutter the figure. Please highlight the features of import only. Also, the text box "threading to permit..." points to a slot which clearly cannot be threaded. Please correct.

Response 12: The intent of the figure was to demonstrate the specifications of the engineering process for the cleat plate. However, the sizing of the figure has rendered the figure difficult to read, and the authors agree with the reviewer that the figure can be improved. Figure 4 has been updated accordingly. The reviewers hope that the amended figure is acceptable.

Point 13: Line 300: Saddle height is a parameter that has a strong impact on cycling kinematics. Applying a 5 mm change in height uniformly across all subjects is appropriate? By what analysis? The value would seem to be dependent on the stature and proportions of each rider. How can you remove the impact of saddle height changes from the data? Have experiment 3 with POS cleat at original saddle and experiment 4 with MPJ cleat at lower saddle?

Response 13: To compensate for the effective decrease in leg length with the POS cleat position, the saddle height was positioned 5 mm lower. This was based on prior research (4. Paton & Jardine, 2012; 14. Millour, G et al., 2020). As the reviewer points out, stature and proportions are relative to the rider. This also applies to bicycle dynamics inclusive of seat tube angle, stack and reach and crank length. Each of these parameters could equally impact cycling kinematics. However, as individual bicycle dynamics were out of scope and not part of the original research remit, in order to create a uniform measurement of sorts that provides a degree of consistency when analysing the results, the 5 mm position was selected. While significant effects were observed between the MPJ and POS cleats, the change in RPE was not significant and was found to have a small effect. This suggests that the 5 mm change did not cause undue fatigue or compromise cycling performance. This is also consistent with prior research.

The reviewer makes an interesting point with regard to removing the impact of saddle height. Peer-reviewed literature recommends the use of a 25° to 35° knee angle for injury prevention and in some instances, 109% of inseam for optimal performance. Despite this, previous research has established that these two methods do not produce similar saddle heights [Peveler, W., 2008. Effects of saddle height on economy in cycling. J Strength Cond Res, 22(4), 1355-9]. Saddle height was normalized as best as possible from the centre of the pedal axle to the saddle top, with the pedal at the most distal end, as per prior research [Gregor et al., 1991]. Additional adjustments would likely comprise muscle recruitment magnitudes and therefore performance efficiency. Whilst it is possible to conduct an experiment with the POS cleat at the original (i.e., self-selected) saddle height, this risks altering muscle recruitment patterns, as per Bini et al. [2014] without considering the impact on pedal force efficiency and overall lower limb movement. This also enhances the risk of injury if knee angle exceeds 35°.  The reviewer’s suggested amendments could be achieved in a laboratory environment whereby participants are closely monitored, albeit tethered, to electronic devices. However, in the current study it is not possible to remove the influence of saddle height from the data. If the reviewer is interested, a prior experiment evaluated the differences to saddle height using sensor technology [Evans et al., 2021. Evaluation of Accelerometer-Derived Data in the Context of Cycling Cadence and Saddle Height Changes in Triathlon. Sensors, 21(3), 871]. The authors hope that this explanation satisfies the reviewer.

Point 14: Figure 5 shows the cleat plate on the shoe. Was the increase in effective leg length resultant from the added height of the cleat plate accounted for in setting saddle height?

Response 14: To compensate for the effective decrease in leg length with the posterior cleat position, the saddle height was positioned 5 mm lower in this condition, as per Paton and Jardine [2012] and Millour et al. [2020].

Point 15: Figure 6 again has text boxes obscuring the image and the text of the rightmost box appears to be the truncated as "...integrate gear shift levers" would seem to be the appropriate text.

Response 15: The original wording was ‘integrated gear shifters’ which was obscured. Figure 6 has now been resized to ensure clarity. The authors thank the reviewer.

Point 16: Line 343: The longitudinal acceleration did not detect the cleat position. The cleat position was found to impact the longitudinal acceleration. Were any significant differences in acceleration observed in the other axes? Ahh, I see this in lines 358. What could the reduction of anteroposterior mean?

Response 16: The authors thank the reviewer for this point. Line 343 has been amended to reflect the reviewer’s statement. The reduction in anteroposterior trunk acceleration was discussed on page 12, line 476. In order to assist the reviewer, a copy is presented below:

Page 13, Line 539:

The reasons for greater longitudinal yet reduced anteroposterior acceleration are not wholly understood, but it can be speculated that such a change may produce alterations in general cycling efficiency. In this sense, anteroposterior motion represents the global (gross) movement of the triathlete during cycling. An alteration in upper body position is related with changes in activation of lower limb muscles [7] whilst changes in cadence can optimise or deteriorate gross and physiological efficiency [35].

Point 17: Line 344: Main peak? This seems to be referring to data or figures not available to the reader and is therefore superfluous. Perhaps a figure of an acceleration waveform is valuable to explain the methodology.

Response 17: The authors have clarified the last section to improve simplicity.

Page 9, line 411:

From the accelerometer dataset, cycling cadence changes were annotated during the execution of synchronisation points. This signified the completion of one 5 km cycling lap and cadence condition in order to categorise synchronisation points in the raw data through post hoc analysis. Therefore, in the current dataset cadence changes were observable due to the corresponding synchronisation points. Longitudinal acceleration was used as an initial indicator of a change to participant trunk acceleration during this analysis.

The above has been included in the paper.

The authors believe the inclusion of an additional figure, whilst helpful, is not desirable in this instant given the revised description and the fact that there already exist nine figures in the paper.

Point 18: Line 346: "longitudinal acceleration for the" is needed between mean and POS.

Response 18: The authors thank the reviewer and confirm that this has been amended.

Point 19: Table 3 seems to contradict the data in Table 2, where the mean accelerations are much less than reported in Table 3. Also, Table 3 seems to show that the difference in anteroposterior acceleration (z-axis) is larger as a percentage than longitudinal (x-axis) between the POS and MDJ positions. Please clarify and explain.

Response 19: The authors express thanks to the reviewer. Table 2 displayed the magnitude of mean triaxial trunk acceleration (in m/s²) in two cleat positions in 20 km cycling. Table 3 provided the descriptive data relating to triaxial acceleration magnitudes from each 5 km lap of cycling in the POS and MPJ cleat positions.

The authors have revised the statistical analysis in order that the reviewer has confidence in the results presented. Consequential, Table 2 has been revised.

Point 20: The labels in Figure 7 are nonsense. NTL is used instead of MDJ and 3 of the 4 data sets are labelled POS. Correct please.

Response 20: Figure 7 has been revised with NTL removed. The authors are unsure why NTL has appeared when the correct reference should be MPJ. The authors apologise for this error.

Point 21: Again Figure 8 uses the otherwise unmentioned NTL nomenclature. Correct.

Response 21: The authors confirm that Figure 8 has been amended.

Point 22: Figure 9 uses "smooth" lines between data points. This implies a model of some sort and should be avoided. The data should not have lines between them, as no data is available nor is a prediction based on a model available. Rather, each data point at the check points should include the error bars from the n=7 subjects. Without error bars Figure 9 is useless. Line 410 references variability, and variability can only be assessed with error bars which represent the data variability.

Response 22: The authors thank the reviewer for the feedback. However, the authors do not agree that Figure is ‘useless’. The purpose of Figure 9 was to display the variation in trunk kinematics during running after cycling in both cycle cleat positions in each individual axis (longitudinal, mediolateral and anteroposterior) per 1 km of running. A mathematical model was not mentioned nor promoted in the text.  Despite this, Figure 9 has been revised further to the reviewer’s concerns.

Point 23: Additionally, Line 411 refers to the slope of the difference in mediolateral acceleration. You have no information as to the slope. No data is presented between 4-5km, and while the smoothed line has a slope, your data does not.

Response 23: Please see prior response. The reference to slope has been removed and data clarified.

Point 24: Line 446: What is disproportionate? How do your results apply to an individual athlete?

Response 24: In order to avoid misperception, line 446 has been clarified with the word disproportionate replaced. The original intention was to highlight how a large magnitude of acceleration in one axis could lead to an inverse relationship in the other.  

As stated on line 403, cleat position appeared to influence the longitudinal-anteroposterior relationship, predominantly during the first two laps of cycling.

Relative to applying results to an individual athlete, use of sensor technology demonstrates how widely available mobile technology can be used to quantify kinematics without the need for a fully instrumented gait laboratory. Consequently, the continuous collection of data recorded by a sensor allows for regular analysis during an actual cycle and running environment which subsequently provides information about what and when changes to trunk acceleration occur. For individual triathletes and/or their coaches, accelerometer outputs could be considered when exploring cleat positions and subsequent cuing of longitudinal acceleration when cycling and/or running in dynamic settings. The small sample size in this study was limited due to COVID restrictions, however, it does set the foundation for a larger and more homogenous cohort.

Point 23: Line 455: Your data showed an increase in longitudinal acceleration but also a larger reduction (on a percentage basis) of anteroposterior acceleration when cycling in the POS as compared to the MDJ cleat position. The relative impact of the two differences seem to be ignored in your paper. Please expand.

Response 23:  Cleat position appeared to influence the longitudinal-anteroposterior relationship, predominantly during the first two laps of cycling. This was tested by comparing the relationship during post-hoc analysis. Despite a very large effect size, ANOVA showed that statistical significance was not detected between the longitudinal-anteroposterior magnitudes between MPJ and POS cleat cycling. Figure 7 displayed the changes between the two cleat positions on a per lap/cadence basis.

Point 24: Why not take the next step and calculate the multi-axis COM motion as an energy expenditure and compare the POS and MDJ positions? Motion of the COM is real expenditure. With continuous accelerometer readings this should be simple.

Response 24: The authors acknowledge the reviewer’s comment, And the authors see great value including the metrics of energy expenditure. However, this would change the scope of the paper as aside from exertion, physiological inferences were kept to a minimum.  In actuality, the authors have recently had a paper accepted that relates COM motion with energy expenditure via the metabolic equivalent and caloric consumption. The reason for choosing another journal as it was considered outside the scope of SENSORS.

Point 25: Finally, discussion of anteroposterior is found in 470. This review was conducted in stream of consciousness, as read, so please bear with me.

Response 25:  The authors understand, and can relate entirely.

Point 26: Line 538: Why not continue this train of thought? Can the acceleration data not be extended as a proxy for energy expended based on assumptions of other factors being equal from session 1 to 2 on the kinematics of the athletes? If overall COM motion is reduced with one or the other cleat position, then one or the other is more efficient.

Point 26: The authors are grateful for the thought and consideration given by the reviewer in order to expand the scope of the paper. The authors agree that acceleration data can be used as a proxy for energy expenditure. The reviewer is directed to response 24.

In the current study, running after POS cycling resulted in a quicker finishing time. One can postulate that this would result in improved, or reduced, energy consumption compared with the slower time in running after MPJ cycling. However, as participants would have been running at a higher VO2 threshold despite running at a self-selected pace, overall work rate may have increased. This does suggest a change in metabolic variables. As mentioned in Response 24, the authors have taken this next step and used accelerometer data against metabolic measures in a forthcoming publication. The task of using accelerometry in cycling and running, particularly when manipulating geometric bicycle settings and observing changes to running, remains an ongoing focus for the authors.

Point 27: Line 577: Please clarify that this applies to only one accelerometer axis, not COM acceleration as a whole.

Response 27:  The authors confirm that this is correct. The addition of ‘longitudinal acceleration’ has been added.

Reviewer 4 Report

This manuscript examines the effect of cleat position during a 20 km cycle on subsequent 5 km running performance. Specifically, a conventional (MPJ) and more posterior cleat position (POS) are compared. Seven recreational triathletes ran 5 km immediately following a 20 km cycle (prescribed pedaling cadences) with the MPJ cleat position and again a week later with the POS position. Accelerometers worn on the lower back continuously recorded data during both cycling and running trials and subjective ratings of perceived exertion were recorded after each 5 km of the 20 km cycle. Subjects finished the 5 km run in a shorter time following cycling in the POS position with lesser mean longitudinal acceleration and with lesser RPE. The effect of cleat position during cycling on running performance and the mechanisms that may underlie potential advantages are of interest to the scientific and sports community. However, this study has several major issues related to the experimental protocol and data analysis.

Major concerns:

Was the order of cleat positions randomized (e.g., was the MPJ position always examined first on day 1 with POS examined on day 2 or was POS ever examined first)? If conditions were not randomized, how do you know your comparison of positions is not corrupted by learning effects? For example, the changes in performance attributed to the POS position may instead be due to increased familiarity with the course/protocol given that the subjects had already completed experiment 1. The authors do note some decisions were made to mitigate within-lap learning effects due to new cadence prescriptions (line 250-251), but not for learning effects between experiments.

The authors propose accelerometer-based metrics, 5 km time, and RPE to understand the effects of the experimental conditions on running performance. In my opinion, the 5 km time and RPE results are the most insightful in the current study as they add to the scarce and varied results that have been published on the matter (e.g., see lines 87-102). Beyond this, the accelerometer-based analysis is the primary novelty of the current study. Several issues concern this analysis:

  1. It is not clear exactly what metrics were computed and how they were computed from the accelerometer data to characterize movement during the cycling and running trials. It is not until the results section that the reader learns mean acceleration was used.
  2. It is not clear why the chosen metrics are appropriate for elucidating any mechanism that may underlie observed differences in running performance between the two conditions. Vertical oscillation of the COM is a position measure and requires double time-integration of the accelerometer signal after correcting for orientation differences between the sensor and the world frame (which was apparently not done). Moreover, component-wise mean acceleration is not an appropriate surrogate. For example, one can show that the mean accelerometer output along the vertical direction of the world frame (i.e., aligned with gravity) over a complete stride cycle should be about 1g (deviations will be due to differences in the initial and final vertical velocities). The values in the current study are less than 1g (e.g., see Table 5, x-direction) because the longitudinal sensor axis is not exactly aligned with the world frame vertical during the run. If, however, mean acceleration were computed for the stance phase only, it could be related to the vertical impulse of the COM and hence vertical oscillation. This information is lost by averaging across all strides which includes stance and flight phases and, likewise, confounds the analysis in the current study.
  3. Accelerometer-based running analysis has been done across many studies in the last two decades. Simple event detection algorithms would have enabled the computation of spatiotemporal metrics (contact time, stride frequency, etc.) that are far more common than mean acceleration. Why were these not assessed?

Specific Comments:

Lines 24-25: “longitudinal motion” can be interpreted several ways. Also, the reported p-value corresponds to the magnitude comparisons (Table 2), not component-wise values, but its placement here seems to associate it with the longitudinal component.

Lines 54-56: “kinematical magnitude of trunk motion” is unclear. Presumably you refer to magnitude of lumbar acceleration since this is what was measured. It is not clear how the text and referenced research preceding this statement suggests lesser lumbar acceleration magnitude is advantageous in cycling.

Lines 87-102: two studies are referenced in the first sentence of the paragraph (4 and 14). However, I see only references 4 and 15 discussed in the paragraph.

Lines 110-111: interventions for what? Is this referring to running or cycling (both are discussed in the paragraph)? The effect of trunk position on performance is supported, but it is not clear how acceleration magnitude provides the same insight. It seems one could observe larger trunk acceleration magnitudes simultaneously with optimal positioning of the trunk; they are not mutually exclusive. The claim that lesser trunk acceleration magnitude is beneficial seems unsupported from the previous literature (acceleration is different than position). Perhaps one could argue it is more metabolically demanding. However, your results do not support this: despite a significant increase in trunk acceleration magnitude there was no difference in RPE (Table 3).

Line 115: does oscillation refer to displacement? Vertical displacement and acceleration are different. Moreover, the latter may not be an appropriate surrogate for the former without time duration information.

Lines 141-152: why was a control run not performed where the subject runs 5 km without having just completed a 20 km cycle? This would have allowed an understanding of how running following cycling in a particular cleat position differs from the optimal scenario (i.e., unfatigued, not following cycling).

Lines 166-167: what is the point in defining delta_x = x_f – x_0? It is not used elsewhere.

Lines 190-194: how do you know participants stayed within the prescribed cadence range? Why was cadence prescribed instead of speed? Couldn’t two subjects pedal with the same cadence and expend different amounts of energy (e.g., depending on the gear)? If so, this introduces an extraneous variable that could further confound the comparison between the two conditions. Reporting the 20 km cycle time (even better would be the individual lap times) in the two conditions would help to clarify.

Lines 210-211: “The precise location was selected …” It is unclear what is being conveyed in this sentence.

Lines 216-219: again, based on the previous research that has been cited, it seems undesirable trunk movement would refer to trunk position (see comment on lines 110-111); not necessarily acceleration-related variables.

Lines 217-218: might subtle movement in the frontal plane (which would excite the medio-lateral sense axes of the accelerometer) be advantageous if it results in a positioning of the lower limb that allows generation of larger applied forces to the pedal in the drive phase?

Lines 223-224: what sensor range was selected?

Line 248: it is unclear what is meant by, “…accelerations of each component were likened for each participant…”

Line 251: what drift effect was concerning? Drift is a phenomenon resulting from time-integration of a biased signal.

Line 260-263: this may be true during the stance phase, but the relation between average acceleration and vertical displacement is lost when averaging over a complete stride cycle.

Line 264: “running lap.” Figure 2 suggests running and cycling were performed on the same 5 km loop. Therefore, didn’t subjects run only one “lap”?

Lines 318-319: was RPE assessed during the running trials as well?

Line 332: exactly what variables were compared between the two conditions? Why not do a paired-samples t-test (assuming there was one variable per subject to characterize each condition)?

Line 337: correlation coefficients were computed? For what pairs of variables?

Lines 338-340: “The root mean square (RMS) values…” This sentence seems out of place and may be better placed in the methods section where all outcome variables are described. What is an “RMS proportional deviation in acceleration magnitude”?

Lines 342-344: last sentence of section 2.5 needs clarification. A figure may help.

Line 346: it should be clarified that the data reported here are accelerations. Was this across the entire 20 km cycle?

Line 353: is “feasible” meant instead of “conceivable”?

Line 355: effect sizes are not in Table 3, but the text says they are.

Line 355-358: significantly larger longitudinal acceleration is reported in the main text for the final lap as if it wasn’t for any other lap. But Table 3 suggests the same was observed for every lap. Why is the insignificant larger RPE (should be clarified that it was larger for MPJ) reported in the main text for the final lap but not the insignificant larger RPE for POS in laps 1 and 2?

Line 372: where is d defined? Line 337 seems to suggest r was used as an effect size.

Lines 379-380: the acceleration magnitudes in running seem low (Table 4); especially compared to the cycling magnitudes. Figures displaying some representative time-series data would be insightful, especially since investigation of trunk-worn accelerometer signals in cycling are not very common.

Lines 388-389: where was the foot-strike detection analysis described? The text here makes it sound like figure 8 displays acceleration magnitudes at foot strike, but the figure 8 caption refers to cumulative triaxial accelerations. This is unclear. Also, are these acceleration magnitudes in figure 8? If so, why are they so much larger than the acceleration magnitudes reported in Table 4?

Lines 399-400: the text refers to effect sizes in Table 5, but there are none in Table 5.

Line 501: what is cumulative trunk acceleration? All outcome variables used for evaluation should be defined in the methods section and their computation clearly defined.

Line 551-553: how do you know the contact times for the runners in your study were affected by previous cycling?

Table 4: please check the effect size for RPE. Should this be 0.9 instead of 0.1?

Figure 4: image text is blurry and difficult to read.

Figure 7: what is NTL? Also, the labels for the dark and light blue squares are the same.

Figure 8: what is NTL?

Figure 9: is this the same data in Table 5? If so, only one will suffice. In my opinion, the figure is preferred.

Author Response

Reviewer 4

Point: Was the order of cleat positions randomized (e.g., was the MPJ position always examined first on day 1 with POS examined on day 2 or was POS ever examined first)? If conditions were not randomized, how do you know your comparison of positions is not corrupted by learning effects? For example, the changes in performance attributed to the POS position may instead be due to increased familiarity with the course/protocol given that the subjects had already completed experiment 1. The authors do note some decisions were made to mitigate within-lap learning effects due to new cadence prescriptions (line 250- 251), but not for learning effects between experiments.

Response:  The authors thank the reviewer for the considered responses. Cleat position was not randomized in order to maintain consistency with the triathletes current training programs. Additionally, this study followed similar protocols in that one cleat position followed by running was tested prior to another, thus with no randomization (Millour et at.,2020; Paton & Jardine, 2012).

Whilst participants used their own personal triathlon bicycles and were familiar with the overground cycling and running course, they were not familiar with cycling in a POS position nor running after POS cycling. A duration of 7 days separated experiment 1 (day 1) and experiment 2 (day 2) (line 194) to allow adequate recovery and to reduce possible learning effects. The cadence conditions used in this study were verbally communicated to participants prior to both experiments, and again upon completion of each 5 km cycling lap (i.e., when a change of cadence was required). Participants were not given information regarding these cadence conditions during the seven days between tests nor were they informed of the actual POS measurement. Additionally, participants were not informed that the cadence conditions would be the same or in the same order. Thus, participants were merely informed that they would be cycling in four different cadence conditions in both experiments.  In this instance, the authors belief that any possible learning effects would be minimal given the experience of the participants in MPJ cycling and running after MPJ cycling. Despite this, the authors acknowledge that learning effects are problematic to define and cannot rule out this possible influence.  Therefore, to recognise the reviewer’s concern the authors have revised the following sentences.

Page 4 Lines 194: A duration of 7 days separated experiment 1 (day 1) and experiment 2 (day 2) in order to reduce the possible influence of learning effects. Participants were not provided with the cadence conditions between this period and were not informed if they same cadences would be used during both experiments.

Page 15, Line 637:  The current study is not without limitations including small sample size and lack of an elite population. Future studies should look to assess full kinematic, kinetic, and loading rate parameters associated with changes in spatiotemporal measures in both cycling and running when cycle geometry is altered. Future studies should also look to see long-term outcomes for changes in metabolic demand. Furthermore, the training age, experience, physiological capability and the possibility of learning effects may have influenced pedalling dynamics and therefore running performance.

Point: The authors propose accelerometer-based metrics, 5 km time, and RPE to understand the effects of the experimental conditions on running performance. In my opinion, the 5 km time and RPE results are the most insightful in the current study as they add to the scarce and varied results that have been published on the matter (e.g., see lines 87-102). Beyond this, the accelerometer-based analysis is the primary novelty of the current study.

Response: The authors thank the reviewer for their opinion.

Several issues concern this analysis:

Point 1. It is not clear exactly what metrics were computed and how they were computed from the accelerometer data to characterize movement during the cycling and running trials. It is not until the results section that the reader learns mean acceleration was used.

Response 1:  The authors thank the reviewer, although we assume that the reviewer is suggesting that the metrics (i.e., mean acceleration) are moved to the methods section whilst being made clearer for readers. Metrics were defined in the original paper; however, the authors have subsequently revised the methods section to state:

Page 4, Line 160: Cycling and running kinematics data were assessed by means of triaxial accelerometer outputs. Specifically, to measure trunk acceleration magnitude changes to cycling cadence in both cleat positions, and acceleration magnitudes in running after cycling, expressed in m/s².  Data for each cycling variable (cadence) and cleat position were registered during the overground cycle and were calculated as a 60 second mean of each 5 km lap of the overground bicycle course. The purpose of this was to ensure that a stable pacing strategy and cadence stabilisation was attained. Next, to analyse conceivable kinematical variation between MPJ and POS cleat positions during each 5 km lap of cycling and corresponding cadence, mean data were stratified into x, y, z components. Ensuing triaxial acceleration of the trunk in running was calculated as the average magnitude for each 1 km epoch running. Mean running data were also stratified into x, y, z components relative to each 1 km epoch of running.

Point 2: It is not clear why the chosen metrics are appropriate for elucidating any mechanism that may underlie observed differences in running performance between the two conditions. Vertical oscillation of the COM is a position measure and requires double time-integration of the accelerometer signal after correcting for orientation differences between the sensor and the world frame (which was apparently not done). Moreover, component-wise mean acceleration is not an appropriate surrogate. For example, one can show that the mean accelerometer output along the vertical direction of the world frame (i.e., aligned with gravity) over a complete stride cycle should be about 1g (deviations will be due to differences in the initial and final vertical velocities). The values in the current study are less than 1g (e.g., see Table 5, x-direction) because the longitudinal sensor axis is not exactly aligned with the world frame vertical during the run. If, however, mean acceleration were computed for the stance phase only, it could be related to the vertical impulse of the COM and hence vertical oscillation. This information is lost by averaging across all strides which includes stance and flight phases and, likewise, confounds the analysis in the current study.

Response 2:  The authors disagree with this comment. The study looked at temporal trunk kinematics during cycling based on changes to cleat position and not purely running. By studying, in detail, how the relationship between trunk acceleration, cleat position, cadence and the consequential effect on running, more insight may be obtained in how these variables are related. Consequently, the purpose of this study was twofold. First, to characterise the change in trunk acceleration magnitude in triathletes instructed to cycle with an MPJ cleat position and posterior (POS) cleat position. Second, to determine if a POS cleat is more beneficial due to lower magnitudes of longitudinal acceleration when running from cycling

Studies comparing the biomechanical characteristics of elite and good runners found that elite distance runners have slightly less vertical oscillation than good runners [Cavagna GA, Heglund NC, Willems PA. Effect of an increase in gravity on the power output and the rebound of the body in human running. J Exp Biol. 2005;208, 2333–46; Tartaruga MP et al. The relationship between running economy and biomechanical variables in distance runners. Res Q Exerc Sport. 2012;83(3):367–75; Cavanagh PR, Pollock ML, Landa J. A biomechanical comparison of elite and good distance runners. Ann N Y Acad Sci. 1977;301:328–45]. Similarly, Williams and Cavanagh [91] showed a trend, although nonsignificant, towards less vertical oscillation. The intuitive perception is that vertical oscillation is adversely related to economy.  Taking a mean acceleration of the COM across all strides is not uncommon and has been used in running [Lin, S. P., Sung, W. H., Kuo, F. C., Kuo, T. B., & Chen, J. J. (2014). Impact of Center-of-Mass Acceleration on the Performance of Ultramarathon Runners. Journal of human kinetics] and walking [Fazio P et al. Gait measures with a triaxial accelerometer among patients with neurological impairment. Neurol Sci, 34:1–6] when an accelerometer is used to measure the instant accelerations of the three axial components.

Additionally, the authors wish to state that temporal kinematics of the trunk in running also highlighted the variations to mediolateral and anteroposterior motion and not only longitudinal acceleration. Although minimal differences were seen in both mediolateral and anteroposterior motion aside from the second km of running, the average 5 km run time was significantly faster in running after POS cycling. Vertical acceleration of the trunk was significantly lower in POS running, however, that alone may not be the biggest contributor given the differences to the root mean square relative to mediolateral accelerations of the trunk.

The reference frame was considered in calculating raw acceleration data with conversation factors and accelerations appropriately scaled into m/s². The authors wish to highlight that the purpose of the study was not to compute mean accelerations in the stance phase to use as a proxy for acceleration magnitudes when considering additional factors such as impulse and ground reaction force. Instead, the study evaluated the effectiveness of a triaxial accelerometer to determine acceleration magnitudes of the trunk in outdoor cycling in two different bicycle cleat positions and the consequential impact on trunk acceleration during running. In this sense, temporal trunk kinematics and associated magnitudes was the metric and mechanism used to infer such changes in both cycling and running.

Point 3. Accelerometer-based running analysis has been done across many studies in the last two decades. Simple event detection algorithms would have enabled the computation of spatiotemporal metrics (contact time, stride frequency, etc.) that are far more common than mean acceleration. Why were these not assessed?

Response 3: The authors acknowledge the reviewer’s comment in that spatiotemporal metrics in running are frequently reported in the literature. However, the focus of this research was to examine trunk acceleration magnitudes in triathlon cycling and running and not spatiotemporal metrics. Prior research has suggested that these oscillations and/or accelerations may be a trainable parameter and that efficient runners display lower magnitudes of oscillations. Furthermore, mean acceleration is both a measurable and potential indicator of running performance [e.g., Lee et al. (2009). Identifying symmetry in running gait using a single inertial sensor, jsams]. While a detection algorithm could have been written, the methodology was deliberately kept relatively simple to allow sports scientists, athletes and coaches without coding and/or programming experience to replicate the study and use acceleration data in the field.  In the authors’ opinion, the inclusion of algorithms, whilst simple for engineers and those familiar with computation techniques, would have added a layer of complexity to the study design. That is not discounting the importance of contact time, stride frequency, stride rate and stride length, and it is the authors’ intention to include these metrics in future studies.

Specific Comments:

Point 4: Lines 24-25: “longitudinal motion” can be interpreted several ways. Also, the reported p-value corresponds to the magnitude comparisons (Table 2), not component-wise values, but its placement here seems to associate it with the longitudinal component.

Response 4:  The authors thank the reviewer for highlighting this. The authors agree that longitudinal motion could be construed differently. Reference to the p-value has been clarified.  Therefore, this has been revised and now reads:

Abstract: The evaluation of accelerometer derived data within a characteristic overground setting suggests a significant increase in total trunk acceleration magnitude (p = 0.04) despite a small effect (d = 0.2) to the ratings of perceived exertion (RPE).

Point 5: Lines 54-56: “kinematical magnitude of trunk motion” is unclear. Presumably you refer to magnitude of lumbar acceleration since this is what was measured. It is not clear how the text and referenced research preceding this statement suggests lesser lumbar acceleration magnitude is advantageous in cycling.

Response 5. Longitudinal acceleration can be described as the amount of displacement that the body centre of mass (COM) experiences (i.e., the up and down motion of the COM). Since a substantive proportion of triathlon race duration is spent cycling and running, it is important to recognise the significance of trunk position given the abrupt change from a near horizontal trunk profile in cycling to a vertical trunk in running. In this regard, trunk angle affects leg kinematics, related muscle activity and limb length (Savelberg et al., 2003. Body configuration in cycling affects muscle recruitment and movement pattern. J. App. Biomech, 19, 310–324) with a forward shift of the COM meaning that the athlete is less supported by the saddle.

Past research by Too (1994, Too D. The effect of trunk angle on power production in cycling. Res Q Exerc Sport, 65(4), 308-15) revealed that peak power at the 60° and 90° trunk angle was significantly greater than that at the 120° angle, and mean power in the 90° angle was significantly greater than that at the 120° angle. It was concluded that changes in cycling trunk angle may affect peak power and mean power. Although trunk angle and not acceleration magnitudes were researched in the current paper, it could be argued that a lesser magnitude of acceleration may also affect peak/mean power given the power-cadence relationship.  Nonetheless, the authors agree with the reviewer in that the original reference was unclear and have revised the paragraph to state;

Page 2, Line 51: Since a substantive proportion of race duration is spent cycling and running, it is important to recognise the significance of the trunk given the abrupt change from a near horizontal trunk profile in cycling to a vertical trunk in running. For instance, trunk angle affects leg kinematics, related muscle activity and limb length [7] with a forward shift of the COM meaning that the athlete is less supported by the saddle [9]. It could be argued that a lesser COM displacement may also affect peak/mean power given the power-cadence relationship. This raises questions on both the limiting factors as well as the significant relationship between the body COM in cycling and running.

Point 6: Lines 87-102: two studies are referenced in the first sentence of the paragraph (4 and 14). However, I see only references 4 and 15 discussed in the paragraph.

Response 6:  Reference to study 14 was originally stated on line 59. To improve clarify and ensure consistency, lines 87-102 have been revised to reintroduce study 14. The amended version now reads:

Page 3, Line 104:

Two studies have investigated the effects of a mid-foot cleat position on biomechanical variables during both cycling and running of a simulated sprint distance run.  Millet et al. [2020] [14] evaluated the impact of fore-aft ad POS cleat position on biomechanical and physiological variables during the cycling and running parts of a simulated Sprint triathlon. The authors concluded that the POS cleat position could have practical benefits for subsequent running and could be recommended for use in triathlons when running 5 km and over. Paton and Jardine [4] showed that a mid-foot cleat position improves the subsequent performance of a 5.5 km time trial running exercise after 30 minutes of cycling at 65% of the mean arterial pressure (MAP).

Point 7: Lines 110-111: interventions for what? Is this referring to running or cycling (both are discussed in the paragraph)? The effect of trunk position on performance is supported, but it is not clear how acceleration magnitude provides the same insight. It seems one could observe larger trunk acceleration magnitudes simultaneously with optimal positioning of the trunk; they are not mutually exclusive. The claim that lesser trunk acceleration magnitude is beneficial seems unsupported from the previous literature (acceleration is different than position). Perhaps one could argue it is more metabolically demanding. However, your results do not support this: despite a significant increase in trunk acceleration magnitude there was no difference in RPE (Table 3).

Response 7:  The authors thank the reviewer for providing this constructive feedback. The term intervention was used in reference to possible training improvements. In this sense, where trunk position influences power output, it is reasonable to suggest than a change in trunk position will also change the acceleration component. Along this line, if trunk position is a trainable parameter, then so, too, could the temporal acceleration magnitudes of the trunk. The authors considered this supposition, which is why it was stated that instructing a triathlete to reduce acceleration of the trunk may be a viable alternative to traditional methods of interventions.

The reviewer is correct that trunk position differs from acceleration, although the ideal trunk positioning is relative to the individual athlete and bicycle geometry. One could speculate that a change in upper body posture due to cadence and/or perceptual exertion would result in a greater range to temporal differences in magnitude which would be identifiable in the data. For example, increases to mediolateral acceleration during cycling could be indicative of mediolateral sway despite the cycling having an ‘optimal’ trunk position. Mediolateral sway and associated magnitudes could likely be considered inefficient due to the linear direction of travel.

Interestingly, as the reviewer states, the argument could be due to metabolic changes, particularly if future studies examined cycling at higher intensities or for longer duration. Whilst a significant increase in trunk acceleration was observed for cycling in a POS cleat position despite an insignificant change to RPE, this was not observed in running after POS cycling. In this instance, RPE was significantly different with extremely large effects (Table 4), suggesting a lessening of metabolic demand. Notwithstanding these arguments, and as the reviewer correctly cautions, the unsupported literature does not necessarily support these findings. However, it is important to note that the literature does not necessarily dispute these findings either. This is largely due to the lack of empirical research on using accelerometry in a simulated cycle to run study, specifically when looking at the impact of cycling cleats and the impact of running. This highlight both the novelty and limitations of the current study.

Point 8: Line 115: does oscillation refer to displacement? Vertical displacement and acceleration are different. Moreover, the latter may not be an appropriate surrogate for the former without time duration information.

Response 8: The authors thank the reviewer for raising this point. In this instance. Displacement can relate to a change in a position vector. In an oscillatory motion, displacement simply means a change in any physical property with time. Therefore, vertical oscillation is the amount of displacement up and down during running. Further, the change in patterns to mediolateral and anteroposterior acceleration magnitudes can also be used as indicators of efficiency [i.e., Lee et al. 2009). Identifying symmetry in running gait using a single inertial sensor, jsams]. Whilst vertical displacement and acceleration differ, this paper studies the impact that cleat position had on triaxial accelerations of the trunk in cycling and running on an individual’s natural cycling and running position and studies how cleat position could be used to counteract this effect.

Point 9: Lines 141-152: why was a control run not performed where the subject runs 5 km without having just completed a 20 km cycle? This would have allowed an understanding of how running following cycling in a particular cleat position differs from the optimal scenario (i.e., unfatigued, not following cycling).

Response 9: The focus of the current study was to explore differences between cycling in different cleat positions and the consequential impact on running in these cleat positions. A common characteristic of triathlon training involves performing cycling and running sessions, known as brick sessions. In this regard, previous cycling may impair running performance in triathlons, so brick training becomes an important part of training [Olcina et al., 2019. Effects of Cycling on subsequent running performance, stride length, and muscle oxygen saturation in triathletes. Sports (Basel, Switzerland), 7(5), 115]. The principle of specificity suggests that since this skill is a critical transition in a triathlon, having further impact on the subsequent running section, practicing this skill is vital for success [Hamworth et al., 2010. Training for the bike to run transition in triathlon].  As the study design was conducted overground on a representative cycling and running route, the focus was to simulate, as close as possible, a characteristic training session. In this regard, running immediately after cycling is common in triathlon training compared with duathlon which involves running pre and post cycling. All triathletes in the current study had previously cycled using the MPJ cleat position and were well experienced in running immediately afterward. In this regard, running post MPJ cycling can be considered as a baseline measurement of sorts given the experience of triathletes.  The authors believe that the influence of fatigue, or perceptual exertion via the RPE method, during cycling was minimal based on participant responses. In this sense RPE did not exceed 13 (described as somewhat hard) [Borg, 1998] and therefore fatigue could be considered as having negligible influence. However, the authors concede that the omission of a control run could cause confusion and have therefore included the following:

Page 4, Line 172

2.1. Methodology

Triathletes include workouts in their training plans that stack two disciplines, one after the other, with minimal to no breaks in between. This is because one of the most important aspects of this sport is the transition from cycling to running, which is a key factor in achieving a good result [1, 2]. Therefore, to replicate a typical training condition and to accomplish the purpose of this study, participants cycled at a varied yet progressively augmented cadence for 20 km followed by a 5 km overground run whilst wearing a triaxial accelerometer.

The authors thank the reviewer and hope that this meets expectations.

Point 10: Lines 166-167: what is the point in defining delta_x = x_f – x_0? It is not used elsewhere.
Response 10:  The displacement definition was used to signify to the reader that triathletes commenced and finished both cycling and running experiments at the same location. This was to complement Figure 2 (the overground cycle route). Nevertheless, the authors have removed this reference.
Point 11: Lines 190-194: how do you know participants stayed within the prescribed cadence range? Why was cadence prescribed instead of speed? Couldn’t two subjects pedal with the same cadence and expend different amounts of energy (e.g., depending on the gear)? If so, this introduces an extraneous variable that could further confound the comparison between the two conditions. Reporting the 20 km cycle time (even better would be the individual lap times) in the two conditions would help to clarify.
Response 11: It is possible that participants pedal at the same cadence and expend different amount of energy, however, this also applies to running and many other sporting codes. This extraneous variable was originally mentioned on page 15, line 612 as a limitation of the study. As is common in kinematic studies conducted in the field, monitoring energy expenditure or maximum oxygen update is not always possible or feasible. Additionally, this was why a previously used (and accepted in peer review) cadence protocol and range was used [Chapman et al. 20A protocol for measuring the direct effect of cycling on neuromuscular control of running in triathletes, J Sport Sci, 27, 767–782]. Each triathlete cycled with cadence meters attached to the bicycles. Cadence was viewable via the individual display meters in order that participants could monitor the appropriate rev/min¹. Data from these cadence meters was able to be download and analysed for each athlete to obtain an approximation of their average cadence per 5 km lap. Nevertheless, the authors acknowledge the reviewer’s suggestion concerning the reporting of 20 km cycle time and have therefore included the mean 5 km cycle time. The author is directed to Table 3 to view the mean times between MPJ and POS cycling.

Response 12: Lines 210-211: “The precise location was selected …” It is unclear what is being conveyed in this sentence.

Response 12: The precise location referenced relates to the L5/S1 placement of the accelerometer onto each participant. This in line with prior research [Gleadhill S, Lee JB James D. The development and validation of using inertial sensors to monitor postural change in resistance exercise. Journal of Biomechanics, 2016, 3;49(7):1259-1263; Faber GS, Kingma I, Bruijn SM, van Dieën JH. Optimal inertial sensor location for ambulatory measurement of trunk inclination. Journal of Biomechanics, 2009; 42: 2406-2409; Gleadhill et al., 2018. Validating Temporal Motion Kinematics from Clothing Attached Inertial Sensors, Computer Science]

Point 13: Lines 216-219: again, based on the previous research that has been cited, it seems undesirable trunk movement would refer to trunk position (see comment on lines 110-111); not necessarily acceleration elated variables.
Response 13:  The reviewer is directed to response 7.

Point 14: Lines 217-218: might subtle movement in the frontal plane (which would excite the medio-lateral sense axes of the accelerometer) be advantageous if it results in a positioning of the lower limb that allows generation of larger applied forces to the pedal in the drive phase?
Response 14:  The reviewer makes an interesting point, of which the authors are appreciative. In this instance the subtlety of frontal plane movement would arguably need to be defined as frontal plane movement is a deviation from the predominate motion that is performed in the sagittal plane during cycling. Theoretically, a slight deviation in the frontal plane could be advantageous when considering the power phase during the pedal cycle. This would inevitably depend on where in the downstroke, or at which point from the top dead centre (TDC,0°) to the bottom dead centre (BDC, 180°) the frontal plane movement occurred and how the resultant force was applied tangential to the pedal. Furthermore, the relative magnitude of excitement of the mediolateral axis would need to be considered as too high a magnitude could compromise possible advantageous movement during the drive phase. Interestingly, frontal plane movement does exist in cycling, albeit during knee movements. During the power phase, the knee adducts as it extends. This motion leads to medial translation of the knee while the knee extends. By changing trunk movement and altering the magnitude of acceleration, a possible outcome could result in increased trunk sway when working at higher workloads (or cadences), which may increase hip and therefore knee extension movement in the frontal plane. To the authors’ knowledge, this area of research has yet to be explored, although one can only postulate as to what would, or could, occur. This could be an area for future research.

Point 15: Lines 223-224: what sensor range was selected?
Response 15:  The selected sensor range was +16 g. Confirmation of this range has been added to line 265.

Point 16: Line 248: it is unclear what is meant by, “…accelerations of each component were likened for each participant…”
Response 16: The authors acknowledge that the term ‘likened’ is ambiguous. This has been amended to read:

Page 7, line 309:

“… accelerations of each component were collected for each participant…”

Point 17: Line 251: what drift effect was concerning? Drift is a phenomenon resulting from time-integration of a biased signal.
Response 17: Reference to drift was specified based on prior submissions and feedback relative to manuscripts submitted to this journal. Whilst the authors believe the influence of drift was minimal, it was considered worth mentioning in order to pre-empt possible questions. The authors are happy to remove this reference if the reviewer believes it is superfluous.

Point 18: Line 260-263: this may be true during the stance phase, but the relation between average acceleration and vertical displacement is lost when averaging over a complete stride cycle.
Response 18: The authors thank the reviewer for this comment. Whilst the authors note this comment, overall running kinematics after cycling might be impaired with a significant decrease in terms of stride length, which significantly reduces after cycling transition and would therefore alter the stride cycle and mean acceleration over a complete cycle.  In fact, this worsening stride length after cycling has been shown to mainly occur in low-level triathletes [Díaz et al., 2012. Longitudinal changes in response to a cycle-run field test of young male national "talent identification" and senior elite triathlon squads. J Strength Cond Res, 26(8):2209-19] which were the cohorts used in the current paper.  Changes in acceleration and displacement have been associated with changes in ground contact time and stride length, as well as peak knee flexion and the magnitude of lower limb muscle activation, all of which were out of scope of the current study. Past research supports the general notion that reduction vertical acceleration is beneficial relative to running economy and efficiency.

Point 19: Line 264: “running lap.” Figure 2 suggests running and cycling were performed on the same 5 km loop. Therefore, didn’t subjects run only one “lap”?
Point 19: The reviewer is correct in that running and cycling were performed on the same 5 km loop, as stated in Figure 2. The authors acknowledge that the original sentence on line 264 could cause confusion and have revised accordingly. It now reads:

Page 7 Line 325

Triaxial trunk acceleration in running was analysed for 60 seconds at the end of each completed 1 km epoch (i.e., a total of 5 km which represented a complete loop).

Point 20: Lines 318-319: was RPE assessed during the running trials as well?
Response 20:  To ensure consistency with the cycling experiment, RPE in running was assessed at the end of the 5 km run (Table 4). Though, lines 318-319 have been amended to clarify the measure. It now reads:

Page 8, line 387

Perceptual exertion was described by participants upon completion of each 5 km cycle lap and at the conclusion of 5 km running during both experiments using the Borg 6–20 rating of perceived exertion (RPE) scale [32].

Point 21: Line 332: exactly what variables were compared between the two conditions? Why not do a paired samples t-test (assuming there was one variable per subject to characterize each condition)?
Response 21: For convenience, a two-factor ANOVA with repeated measures was used rather than repeated paired sampled t-tests. In particular, a two factor (cleat position x cadence) repeated measures AVOVA to statistically compare cycling relative to triaxial acceleration/temporal trunk kinematic measures with a repeated measures ANOVA was used to compare triaxial acceleration/temporal trunk kinematic measures in both cleat conditions. Ratings of perceived exertion (RPE) was implemented as a covariate.

Point 22: Line 337: correlation coefficients were computed? For what pairs of variables?
Response 22: The authors have expanded on this as it relates to point 29 raised by the author. The effect size statistical test was used to represent the absolute value of the correlation coefficient an effect size in order to summarise the strength of the relationship between triaxial accelerations during MPJ and POS cycling and running after both conditions. As the Cohen's d statistic represents the differences of means expressed in terms of the pooled within group standard deviation (with r being the universal measure of effect size that is a simple function of d), both were used. The authors note that such interchangeability could cause confusion and have classified d as an effect size.  This has now been revised. The authors thank the reviewer.

Point 23: Lines 338-340: “The root mean square (RMS) values…” This sentence seems out of place and may be better placed in the methods section where all outcome variables are described. What is an “RMS proportional deviation in acceleration magnitude”?
Response 23: Further to the reviewer’s suggestion, reference to the RMS has been moved to the methods section. Reference to proportional deviation has been removed. The sentence now reads:

Page 7, line 326

The root mean square (RMS) values in the three sensing axes were then calculated and used as a measure of the magnitude of trunk acceleration.

Point 24: Lines 342-344: last sentence of section 2.5 needs clarification. A figure may help.
Response 24: The authors have clarified the last section to improve simplicity.
Page 9, line 412

From the accelerometer dataset, cycling cadence changes were annotated during the execution of synchronisation points (lines 188-190). This signified the completion of one 5 km cycling lap and cadence condition in order to categorise synchronisation points in the raw data through post hoc analysis. Therefore, in the current dataset cadence changes were observable due to the corresponding synchronisation points. Longitudinal acceleration was used as an initial indicator of a change to participant trunk acceleration during this analysis.

The above has been included in the paper.

The authors believe the inclusion of an additional figure, whilst helpful, is not desirable in this instant given the revised description and the fact that there already exist nine figures in the paper.

Point 25: Line 346: it should be clarified that the data reported here are accelerations. Was this across the entire 20 km cycle?
Response 25: The authors have revised this sentence to include ‘accelerations’. The authors confirm that this applied to the entire 20 km cycle. This clarification has also been added. Point 26: Line 353: is “feasible” meant instead of “conceivable”?

Response 26:  The authors agree with the reviewer and have changed line 426 to reflect this.

Point 27: Line 355: effect sizes are not in Table 3, but the text says they are.
Response 27:  Effect sizes are reported in Tables 2 and 5. The authors thank the reviewer and confirm that this change has been made.

Point 28: Line 355-358: significantly larger longitudinal acceleration is reported in the main text for the final lap as if it wasn’t for any other lap. But Table 3 suggests the same was observed for every lap.
Response 28:  The authors thank the reviewer. The main text references the large increase in longitudinal acceleration for the final lap which also corresponds to the highest cadence (95–100 rev/min¹). While significant differences to longitudinal acceleration were observed between MPJ and POS cycling across all cadence conditions, the final lap was referenced due to this being the largest magnitude observed in both MPJ and POS. The authors have subsequently revised this sentence.

Page 9, Line 427

A significant difference in greater longitudinal acceleration in the POS cleat position was seen during all laps of cycling with the largest magnitudes for both POS and MPJ observed at the highest cadence of 95–100 rev/min¹.

Point 29: Line 372: where is d defined? Line 337 seems to suggest r was used as an effect size.
Response 29: The effect size statistical test was used to represent the absolute value of the correlation coefficient an effect size in order to summarise the strength of the relationship between triaxial accelerations during MPJ and POS cycling and running after both conditions. As the Cohen's d statistic represents the differences of means expressed in terms of the pooled within group standard deviation (with r being the universal measure of effect size that is a simple function of d), both were used. The authors note that such interchangeability could cause confusion and have classified d as an effect size.  This has now been revised. The authors thank the reviewer.

Page 9, Line 406

Threshold values as an effect size (d) were used and classified a 0.1-0.3 (small), >0.3-0.5 (moderate), >0.5-0.7 (large), >0.7-0.9 (very large) and >0.9 (extremely large) [Hopkins et al. 2009. Progressive statistics for studies in sports medicine and exercise science. Med. Sci. Sports. Ex, 41, 3–13].

Point 30: Lines 379-380: the acceleration magnitudes in running seem low (Table 4); especially compared to the cycling magnitudes. Figures displaying some representative time-series data would be insightful, especially since investigation of trunk-worn accelerometer signals in cycling are not very common.

Point 30: The authors express thanks to the reviewer. The accelerometer signals were taken as a mean, as previously mentioned. Given the reviewer’s comments and suggestions, the authors have revised the statistical analysis in order that the reviewer has confidence in the results presented. Consequential, tables and figures have been revised accordingly.

Point 31: Lines 388-389: where was the foot-strike detection analysis described? The text here makes it sound like figure 8 displays acceleration magnitudes at foot strike, but the figure 8 caption refers to cumulative triaxial accelerations. This is unclear. Also, are these acceleration magnitudes in figure 8? If so, why are they so much larger than the acceleration magnitudes reported in Table 4?
Response 31: The reference to sinusoidal arcs and foot strike peaks identified by the accelerometer in running refer to the post hoc analysis completed when reviewing raw accelerometer data. While the word foot-strike detection analysis was not used in the original paper, the authors acknowledge that the sentence could be misconstrued.  As Figure 8 displays triaxial accelerations of the trunk during each 1 km lap of running after cycling in a MPJ and POS cleat position, reference to sinusoidal motion and foot strike peaks has been removed and replaced with:

Page 11, line 461.

Figure 8 shows triaxial acceleration magnitudes in 5 km running after both MPJ and POS cycling with running after POS cycling being of a lower acceleration magnitude.

Point 32: Lines 399-400: the text refers to effect sizes in Table 5, but there are none in Table 5.
Response 32:  The reviewer is directed to point 27.

Point 33: Line 501: what is cumulative trunk acceleration? All outcome variables used for evaluation should be defined in the methods section and their computation clearly defined.
Response 33: The authors once again thank the reviewer. Cumulative trunk acceleration referred to total triaxial acceleration, which was previously reported. The authors have removed reference to cumulative acceleration and instead simply referred to triaxial acceleration magnitudes of the trunk.

Point 34: Line 551-553: how do you know the contact times for the runners in your study were affected by previous cycling?
Response: 34: The authors have revised line 551-553. For the authors to make such a statement, measures of ground contact time would be required. The authors were referring to a prior study that suggested when running after cycling ground contact time and vertical oscillation were not affected by cycling. In the current paper, the authors cannot, and did not, compare ground contact time. In order to clarify, the revised sentence reads:

Page 13, Line 539

The reasons for greater longitudinal yet reduced anteroposterior acceleration are not wholly understood, but it can be speculated that such a change may produce alterations in general cycling efficiency. In this sense, anteroposterior motion represents the global (gross) movement of the triathlete during cycling. An alteration in upper body position is related with changes in activation of lower limb muscles [7] whilst changes in cadence can optimise or deteriorate gross and physiological efficiency [35].

The authors thank the reviewer for raising this issue.

Point 35: Table 4: please check the effect size for RPE. Should this be 0.9 instead of 0.1?

Response 35: The authors agree with the reviewer and have corrected the figure to state 0.9. The SD is not impacted by this change. The authors thank the reviewer.

Point 36: Figure 4: image text is blurry and difficult to read.

Response 36: The intent of the figure was to demonstrate the specifications of the engineering process for the cleat plate. However, the authors agree with the reviewer that the figure can be improved. Figure 4 has been updated accordingly. The reviewers hope that the amended figure is acceptable.

Point 37: Figure 7: what is NTL? Also, the labels for the dark and light blue squares are the same.

Response 37: Figure 7 has been revised with NTL removed. The authors are unsure why NTL has appeared when the correct reference should be MPJ. The authors apologise for this error.

Point 38: Figure 8: what is NTL?
Point 38: Please note the response to the aforementioned point.

Point 39: Figure 9: is this the same data in Table 5? If so, only one will suffice. In my opinion, the figure is preferred.
Response: 39. Table 5 displays descriptive data (means ± SD) from individual 5 km run post cycling in two cleat positions (in m/s2) inclusive of RPE, mean 5 km run completion time and the variation between root mean square. Figure 9 displays a graphical representation of each individual accelerometric axis per 1 km of running (i.e., longitudinal, mediolateral and anteroposterior) without reference to mean completion time, RPE or RMS. The authors aim was to display the data in schematic form to highlight the distinct mean change to trunk acceleration. Notwithstanding this aim, the authors are pleased to accommodate the reviewer’s opinion if, indeed, the figure is more beneficial to the reader.
